# piR-39980 mediates doxorubicin resistance in fibrosarcoma by regulating drug accumulation and DNA repair

Basudeb Das [1], Neha Jain[1] & Bibekanand Mallick [1✉]

Resistance to doxorubicin (DOX) is an obstacle to successful sarcoma treatment and a cause of tumor relapse, with the underlying molecular mechanism still unknown. PIWI-interacting RNAs (piRNAs) have been shown to enhance patient outcomes in cancers. However, there are few or no reports on piRNAs affecting chemotherapy in cancers, including fibrosarcoma. The current study aims to investigate the relationship between piR-39980 and DOX resistance and the underlying mechanisms. We reveal that piR-39980 is less expressed in DOX-resistant HT1080 (HT1080/DOX) fibrosarcoma cells. Our results show that inhibition of piR-39980 in parental HT1080 cells induces DOX resistance by attenuating intracellular DOX accumulation, DOX-induced apoptosis, and anti-proliferative effects. Its overexpression in HT1080/DOX cells, on the other hand, increases DOX sensitivity by promoting intracellular DOX accumulation, DNA damage, and apoptosis. The dual-luciferase reporter assay indicates that piR-39980 negatively regulates *RRM2* and *CYP1A2* via direct binding to their 3′UTRs. Furthermore, overexpressing *RRM2* induces DOX resistance of HT1080 cells by rescuing DOX-induced DNA damage by promoting DNA repair, whereas *CYP1A2* confers resistance by decreasing intracellular DOX accumulation, which piR-39980 restores. This study reveals that piR-39980 could reduce fibrosarcoma resistance to DOX by modulating *RRM2* and *CYP1A2*, implying that piRNA can be used in combination with DOX.

[1]RNAi and Functional Genomics Lab, Department of Life Science, National Institute of Technology Rourkela, Rourkela 769008 Odisha, India.
✉email: mallickb@nitrkl.ac.in

Fibrosarcoma accounts for ~7% of all soft tissue sarcomas and represents about 1% of all solid cancer[1]. The primary treatment of fibrosarcoma consists of amputation, surgical resection of tumor tissues, and radiotherapy[2]. However, these strategies are not beneficial for metastatic or unresectable sarcomas. Approximately 23% of patients with primary extremity sarcoma develop distant metastasis, and the median survival rate is 11.6 months for metastatic patients[3]. As a result, chemotherapy is the most effective alternative option for the comprehensive clinical treatment of sarcoma. Chemotherapy improves disease-free survival, overall survival, and recurrence-free survival from 48 to 80%. Chemotherapy following surgery is effective in treating local recurrence and distance metastasis[4–6].

Doxorubicin (DOX) is considered the first-line chemotherapy drug used to treat soft tissue sarcoma (STS). DOX was initially introduced in the 1970s and remained one of the most effective drugs routinely used in STS treatment[7]. This is an anthracycline drug that halts DNA replication by inhibiting the progression of topoisomerase II after intercalating within the DNA[8]. A recent study showed the response rate of DOX (75 mg/m$^2$) in STS was 14%[9]. The response rate increased to 26% when DOX (75 mg/m$^2$) along with ifosfamide (10 g/m$^2$)[9,10] were given. DOX-based adjuvant chemotherapy significantly reduced local and distant recurrence and increased the overall recurrence-free survival, but there was no improvement in 10 years overall survival[11]. Moreover, response rates to single DOX treatment significantly decrease after exposure to this drug, indicating that growing resistance to DOX can lead to treatment failure. Studies have also shown that DOX exposure induces multidrug resistance (MDR) in sarcoma to daunorubicin, dactinomycin, mitoxantrone, colchicine, vincristine, vinblastine, and etoposide[12].

Several mechanisms can work simultaneously to build resistance against DOX. Most frequent molecular changes that induce DOX resistance include limitation of drug uptake, alteration of drug metabolism, augmented efflux, augmented DNA damage repair, inhibition of apoptosis, and disruption of redox homeostasis[13,14]. However, the exact molecular mechanism of DOX resistance is not fully elucidated in fibrosarcoma. Therefore, it is crucial to further explore the underlying molecular mechanism of DOX resistance in fibrosarcoma and therapeutic strategies to overcome this resistance. Moreover, it is also necessary to find new small molecules that enhance fibrosarcoma cells' sensitivity to DOX and improve therapy clinically.

ncRNA-based therapy is increasingly well accepted as a novel and promising approach in cancer treatment due to advances in delivery strategies. Recent studies have reported that tumor-suppressor miRNA mimics could be systemically delivered using polymer-based vehicles through intravenous injection to inhibit cancer progression[15]. Over the last decade, numerous studies have demonstrated that microRNAs (miRNAs) play key regulatory roles in drug resistance and sensitivity by modulating many molecular events mentioned above[16]. More recently, another class of small ncRNAs termed P-element-induced wimpy testis (PIWI)-interacting RNAs (piRNAs) have emerged as a powerful candidate for cancer therapy[17]. piRNAs are ~23–36 nucleotides (nts) in length and characterized by a 3-terminal 2′-O-methylation with a 5′-terminal uridine or tenth position adenosine bias[18]. piRNAs were initially reported in germline cells, maintaining genome stability by sustaining DNA integrity via silencing of transposable elements[19]. Now, piRNAs are reported in somatic cancer cells regulating proliferation, apoptosis, invasion, migration, angiogenesis, etc[20–26]. However, there are only a few reports on piRNAs playing roles in chemoresistance or sensitivity in cancers, but not in STS or fibrosarcoma. Mai et al.[24] reported that piR-54265 activated the STAT3 signaling pathway,

thereby, induced chemoresistance of colorectal cancer cells[24]. Tan et al.[27] reported piRNA-36712 showed synergistic anticancer effects with two major chemotherapeutic agents, paclitaxel and DOX. Overexpression of piR-36712 significantly decreased the IC$_{50}$ dose of DOX and paclitaxel in MCF7, and ZR75–1 breast cancer cells and mice xenograft models[27]. Roy et al. 2019 has shown overexpression of piR-39980 reduces the sensitivity of DOX and inhibits drug-induced apoptosis in neuroblastoma[28]. However, to the best of our knowledge, there is no study yet on investigating the association of piRNA dysregulations with fibrosarcoma resistance to DOX and implications of piRNA in DOX sensitivity.

Therefore, we aimed to elucidate the role of piR-39980 on DOX sensitivity in HT1080 fibrosarcoma cells and its underlying mechanisms. In our previous study, we have reported that piR-39980 was downregulated in HT1080 fibrosarcoma cells compared to IMR90 normal fibroblast cells and acted as a tumor suppressor by inhibiting proliferation, metastasis, and inducing apoptosis[29]. In this study, we developed HT1080 resistant cell lines to DOX (HT1080/DOX), which exhibited a significantly lower expression of piR-39980 compared to its parental HT1080 cells. The overexpression of this piRNA inhibited Ribonucleotide reductase subunit M2 (RRM2) and Cytochrome P450 1A2 (CYP1A2), two crucial factors for tumor growth and drug resistance. Its overexpression also significantly increased DOX accumulation and promoted drug-induced cell death. This study revealed that piR-39980 could modulate HT1080 cellular response to DOX.

## Results

### piR-39980 is downregulated in DOX-resistant HT1080 fibrosarcoma cells and alters DOX sensitivity

At first, the DOX-resistant HT1080 cell line (HT1080/DOX) was developed from the HT1080 parental cell line by culturing the cells with stepwise increasing DOX concentrations, starting from 10 nM. The HT1080/DOX cell line was established after five subsequent treatments with a 300 nM final DOX concentration. Then, the IC$_{50}$ value of DOX was determined in HT1080 and HT1080/DOX cells by MTT assay. The IC$_{50}$ value of DOX for HT1080 and HT1080/DOX cells were 0.4 μM (Fig. 1a) and 2 μM (Fig. 1b), respectively. The degree of DOX resistance is calculated in terms of resistance index (R), which showed the HT1080/DOX cells were approximately five-fold more resistant to the drug than the parental HT1080 cell line. Further, we found that 0.4 μM DOX (which is IC$_{50}$ for HT1080 cells) can cause only 15% death in HT1080/DOX cells (P = 0.0179, Fig. 1c), whereas 2 μM DOX (which is IC$_{50}$ for HT1080/DOX cells) can cause ~95% death in HT1080 cells (P = 0.0018, Fig. 1d).

Our previous study has shown that piR-39980 is downregulated in HT1080 fibrosarcoma cells compared with IMR90-tert normal fibroblast cells[29]. We performed a qRT-PCR assay again and found the same in HT1080 cells (Supplementary Fig. 1). While comparing the expression of this piRNA in HT1080/DOX cells, we found that piR-39980 is significantly downregulated in these DOX-resistant cells compared to HT1080 parental cells (P = 0.008, Fig. 1e). This tempted us to investigate the effect of piR-39980 on DOX resistance in HT1080 cells. Prior to checking the effect of piR-39980 on the chemosensitivity of HT1080 cells to DOX, the HT1080 cells were transfected with 20 nM piR-39980 mimic and inhibitor followed by qRT-PCR to know the transfection efficiency. The result showed piR-39980 expression was ~260-fold (P < 0.001, Supplementary Fig. 2) increased upon mimic transfection and ~7.5-fold (P < 0.01, Supplementary Fig. 3) decreased upon inhibitor transfection compared to the respective negative controls (NCs).

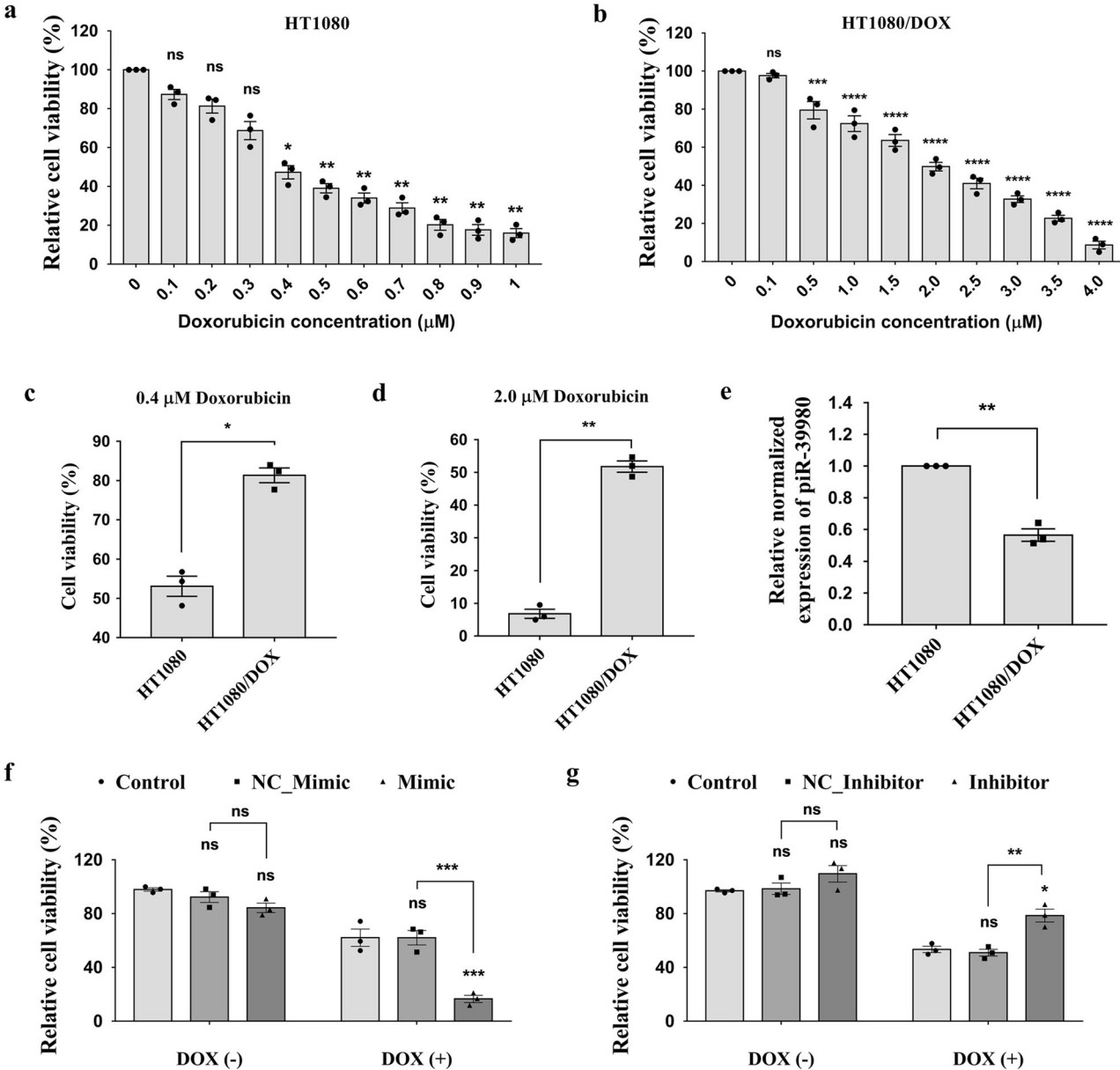

**Fig. 1 The expression of piR-39980 in DOX-sensitive and -resistant HT1080 cells and its effects on cell viability. a** The $IC_{50}$ values of DOX in HT1080 cells determined by MTT assay. Bars, mean ± SEM; $n = 3$ independent experiments; ns nonsignificant, $*P < 0.05$, $**P < 0.01$ vs 0 μM, Dunnett's multiple comparisons test. **b** The $IC_{50}$ values of DOX in HT1080/DOX cells determined by MTT assay. Bars, mean ± SEM; $n = 3$ independent experiments; ns nonsignificant, $***P < 0.001$, $****P < 0.0001$ vs 0 μM, Dunnett's multiple comparisons test. **c** The viability of HT1080/DOX cells in 0.4 μM DOX is ~30% higher than HT1080 cells. Bars, mean ± SEM; $n = 3$ independent experiments; $*P < 0.05$, $t$-test. **d** The viability of HT1080 cells in 2.0 μM DOX is ~40% lower than HT1080/DOX cells. Bars, mean ± SEM; $n = 3$ independent experiments; $**P < 0.01$, $t$-test. **e** Relative expression levels of piR-39980 in parental HT1080 and HT1080/DOX cell lines were determined via qRT-PCR. The expression level of piR-39980 was lower in HT1080/DOX cells compared with HT1080 cells, $p < 0.01$. Bars, mean ± SEM; $n = 3$ independent experiments; $**P < 0.01$, $t$-test. **f** The effect of piR-39980 on viability of HT1080 cells was determined by MTT assay upon transfection with 20 nM piR-39980 mimic/NC_Mimic and treatment with 0.4 μM DOX. Bars, mean ± SEM; $n = 3$ independent experiments; ns nonsignificant, $***P < 0.001$, Sidak's multiple comparisons test. **g** The effect of piR-39980 on viability of HT1080 cells was determined by MTT assay upon transfection with 20 nM piR-39980 inhibitor/NC_inhibitor and treated with 0.4 μM DOX. Bars, mean ± SEM; $n = 3$ independent experiments; ns nonsignificant, $*P < 0.05$, $**P < 0.01$, Sidak's multiple comparisons test.

Initially, the effect of piR-39980 on the chemosensitivity of HT1080 cells to DOX was determined by performing a cell viability assay. Twenty-four hours after transfection, cells were treated with 0.4 μM DOX and incubated for 48 h. We found that relative cell viability was significantly decreased upon mimic transfection than NC_Mimic. The NC_Mimic transfected group showed ~50% viable cells, whereas the mimic transfected group showed ~20% viable cells upon DOX treatment ($P < 0.001$, Fig. 1f). In contrast, relative cell viability was significantly increased upon inhibitor transfection compared to NC_Inhibitor. The NC_Inhibitor transfected group showed ~50% viable cells, whereas the inhibitor transfected group showed ~70% viable cells upon DOX treatment ($p < 0.01$, Fig. 1g). These findings suggest piR-39980 alters the DOX sensitivity of HT1080 cells.

**piR-39980 promotes sensitivity of fibrosarcoma cells to DOX by altering colony formation and apoptosis**. Further, the effect of piR-39980 on DOX sensitivity was also investigated by performing the colony-forming assay. The colony-forming ability of HT1080 parental cells was reduced by 50% upon DOX treatment. Interestingly, DOX reduced the colony-forming ability of this cell to 85% ($P < 0.001$, Fig. 2a) in the presence of piR-39980 mimic, while it reduced to 30% in the presence of piR-39980 inhibitor ($P < 0.01$, Fig. 2b).

Similarly, HT1080 cells transfected with piR-39980 mimic effectively enhanced the DOX-induced apoptosis compared to the cells transfected with NC_Mimic as evident from the apoptosis assay ($P < 0.001$, Fig. 2c). On the contrary, transfection of piR-39980 inhibitor attenuated DOX-induced apoptosis in HT1080 cells compared with NC_Inhibitor ($P < 0.05$, Fig. 2d). These results revealed piR-39980 significantly enhances the sensitivity of HT1080 cells to DOX and hence can be considered a negative regulator of DOX resistance in fibrosarcoma cells.

**piR-39980 increases intracellular DOX-accumulation in parental fibrosarcoma cells**. To investigate the possible cause behind the role of piR-39980 in DOX sensitivity, we performed a DOX-accumulation assay and measured intracellular DOX upon overexpression and silencing of piR-39980 using fluorescence microscopy, multimode microplate plate reader, and flow cytometry. We found an increased level of intracellular DOX by ~3-fold in HT1080 cells transfected with piR-39980 mimic compared to the cells transfected with NC_Mimic ($P < 0.01$, Fig. 3a, b, e). In contrast, intracellular DOX was significantly decreased upon silencing of piR-39980 by its inhibitor ($P < 0.001$, Fig. 3c, d, f). These results indicated that piR-39980 is a positive regulator of DOX accumulation in fibrosarcoma cells.

**piR-39980 induces sensitivity of DOX-resistant fibrosarcoma cells to DOX**. The resistant cells were transfected with piR-39980 to investigate if piR-39980 also influences the chemosensitivity of HT1080/DOX cells. The transfection efficiency of mimic was determined by qRT-PCR, which showed expression of piR-39980 was ~250-fold ($P < 0.05$, Fig. 4a) increased upon mimic transfection compared to NC_Mimic transfected cells.

Initially, we determined the effect of piR-39980 on the chemosensitivity of HT1080/DOX cells to DOX by cell viability (MTT) assay. We found that relative cell viability was significantly decreased upon mimic transfection compared to NC_Mimic. HT1080/DOX cells showed only 15% death in 0.4 μM of DOX, the $IC_{50}$ for HT1080 parental cells. However, we observed 50% death in HT1080/DOX cells upon transfected with piR-39980 mimic along with 0.4 μM DOX ($P < 0.01$, Fig. 4b). Then we checked the effect of piR-39980 on the colony-forming ability of HT1080/DOX cells in the presence of DOX. We found only 20% decrease in colony number in 0.4 μM DOX-treated HT1080/DOX cells compared to untreated cells when cultured for 14 days. However, we found a 50% decrease in colony number when HT1080/DOX cells were transfected with piR-39980 mimic. These results indicated DOX sensitivity of DOX-resistant fibrosarcoma cells was increased by ~2.5-fold due to piR-39980 mimic transfection (Fig. 4c).

We further used piR-39980 mimic and DOX to investigate their influence on HT1080/DOX cell's morphology (Fig. 4d). We noticed that DOX, along with mimic, remarkably induced cell shrinkage and death. Moreover, cells treated with DOX (0.4 μM) showed a little bit better cell morphology than cells transfected with mimic (20 nM) combined with DOX (0.4 μM). The numbers of damaged cells were more in cells treated with mimic along with DOX than untreated cells. These results indicated that combination treatment of piR-39980 mimic and DOX increased cell damage, suggesting a possible synergistic effect of piR-39980 and DOX on anti-proliferation activity.

**piR-39980 increases DOX-accumulation and DOX-induced apoptosis in DOX-resistant HT1080 cells**. We performed fluorescent cell imaging and quantified intracellular DOX upon piR-39980 mimic transfection to see the impact of piRNA on the DOX- accumulation in HT1080/DOX cells. We found a significant increase by ~2.5-fold in intracellular DOX level in HT1080/DOX cells transfected with piR-39980 mimics compared to the cells transfected with NC_Mimic ($P < 0.05$, Fig. 5a). This result confirmed that piR-39980 is a positive regulator of DOX accumulation in DOX-resistant fibrosarcoma cells.

We then performed AO/EB dual staining assay to see the effect of piR-39980 on DOX-induced apoptosis. We found 0.4 μM DOX-induced a lower level of early-stage apoptosis in HT1080/DOX cells (Fig. 5b, white arrow), but noticed an increased level of late-stage apoptotic cells (Fig. 5b, pink arrow) upon mimic transfection (20 nM) combined with DOX (0.4 μM). The PE Annexin-V apoptosis assay even showed an increase in DOX-induced apoptosis in HT1080/DOX cells transfected with piR-39980 mimic compared to the negative control. 20% apoptotic HT1080/DOX cells were observed in the NC_Mimic group treated with 0.4 μM DOX, whereas cells transfected with mimic and 0.4 μM DOX showed 50% apoptotic cells ($P < 0.05$, Fig. 5c).

Hence, we can conclude that restoration of piR-39980 sensitizes the resistant HT1080/DOX cells to the DOX, revealing an essential link of piR-39980 with DOX accumulation and apoptosis induction. Therefore, the loss of piR-39980 plays a crucial role in developing DOX resistance in fibrosarcoma cells.

**piR-39980 promotes DOX-induced DNA damage in the DOX-resistant fibrosarcoma cells**. Seeing the effect of piR-39980 on DOX-induced cell death, we aimed to detect DNA damage, especially DNA double-strand breaks (DSB), which is a crucial feature of apoptosis[30]. First, we performed a single-cell gel electrophoresis assay (termed as comet assay), a sensitive method to detect DNA damage in individual eukaryotic cells[31]. The length of the comet tail and tail moment reflects the number of DNA breaks[32]. We observed a significantly longer tail (~3-fold, $P < 0.05$) in mimic transfected HT1080/DOX cells compared to NC_Mimic transfectant (Fig. 6a and Supplementary Fig. 4). We also found the comet tail moment was ~4-fold ($P < 0.01$) increased in mimic transfected HT1080/DOX cells than NC_Mimic (Fig. 6a and Supplementary Fig. 4). These results indicate piR-39980 induces DOX-mediated DNA damage, which was further validated by detecting phosphorylation of H2AX.

Second, we checked phosphorylation of H2A histone family member X (H2AX) to detect DOX-induced DSBs in HT1080/DOX cells upon transfection with piR-39980 mimic. The phosphorylation of H2AX is an early cellular DNA repair response[33]. Phosphorylated H2AX, termed γH2AX initiates repair mechanisms by establishing an epigenetic signal recognized by downstream repair proteins. This chromatin modification increases the accessibility of DNA for downstream DNA damage response (DDR) proteins, leading to their recruitment and accumulation at break DNA ends. Each break in double-stranded DNA corresponds to one γH2AX focus[33]. We found an increased number of HT1080/DOX cells contain γH2AX foci (number of foci >4) in cells transfected with mimic compared to NC_Mimic (Fig. 6b). More specifically, ~50% of cells in piR-39980 mimic transfected group treated with 0.4 μM DOX showed γH2AX foci in their nucleus. We found ~2-fold increase in γH2AX accumulation in HT1080/DOX cells' nuclei after

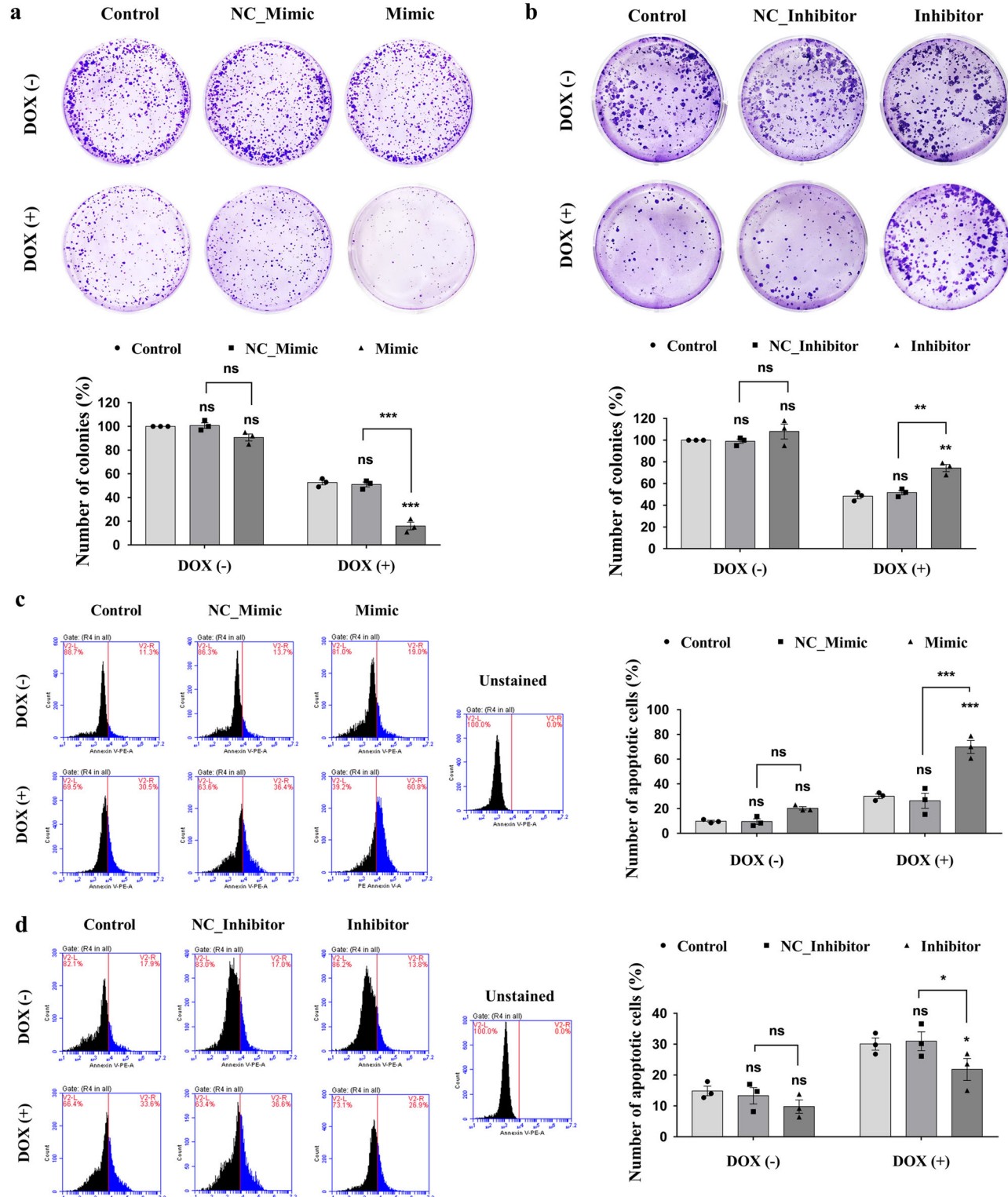

**Fig. 2 Effect of piR-39980 on DOX-mediated colony formation and apoptosis of HT1080 parental cells. a** Colony formation assay after transfecting the cells with 20 nM piR-39980 mimic/NC_Mimic and treatment with 0.4 μM DOX. Bars, mean ± SEM; $n = 3$ independent experiments; ns nonsignificant, ***$P < 0.001$, Sidak's multiple comparisons test. **b** Colony formation assay after transfecting the cells with 20 nM piR-39980 inhibitor/NC_inhibitor and treatment with 0.4 μM DOX. Bars, mean ± SEM; $n = 3$ independent experiments; ns nonsignificant, **$P < 0.01$, Sidak's multiple comparisons test. **c** The apoptosis assay after transfecting the cells with 20 nM piR-39980 mimic/NC_Mimic and treatment with 0.4 μM DOX. Bars, mean ± SEM; $n = 3$ independent experiments; ns nonsignificant, ***$P < 0.001$, Sidak's multiple comparisons test. **d** The apoptosis assay after transfecting the cells with 20 nM piR-39980 inhibitor/NC_inhibitor and treatment with 0.4 μM DOX. Bars, mean ± SEM; $n = 3$ independent experiments; ns nonsignificant, *$P < 0.05$, Sidak's multiple comparisons test.

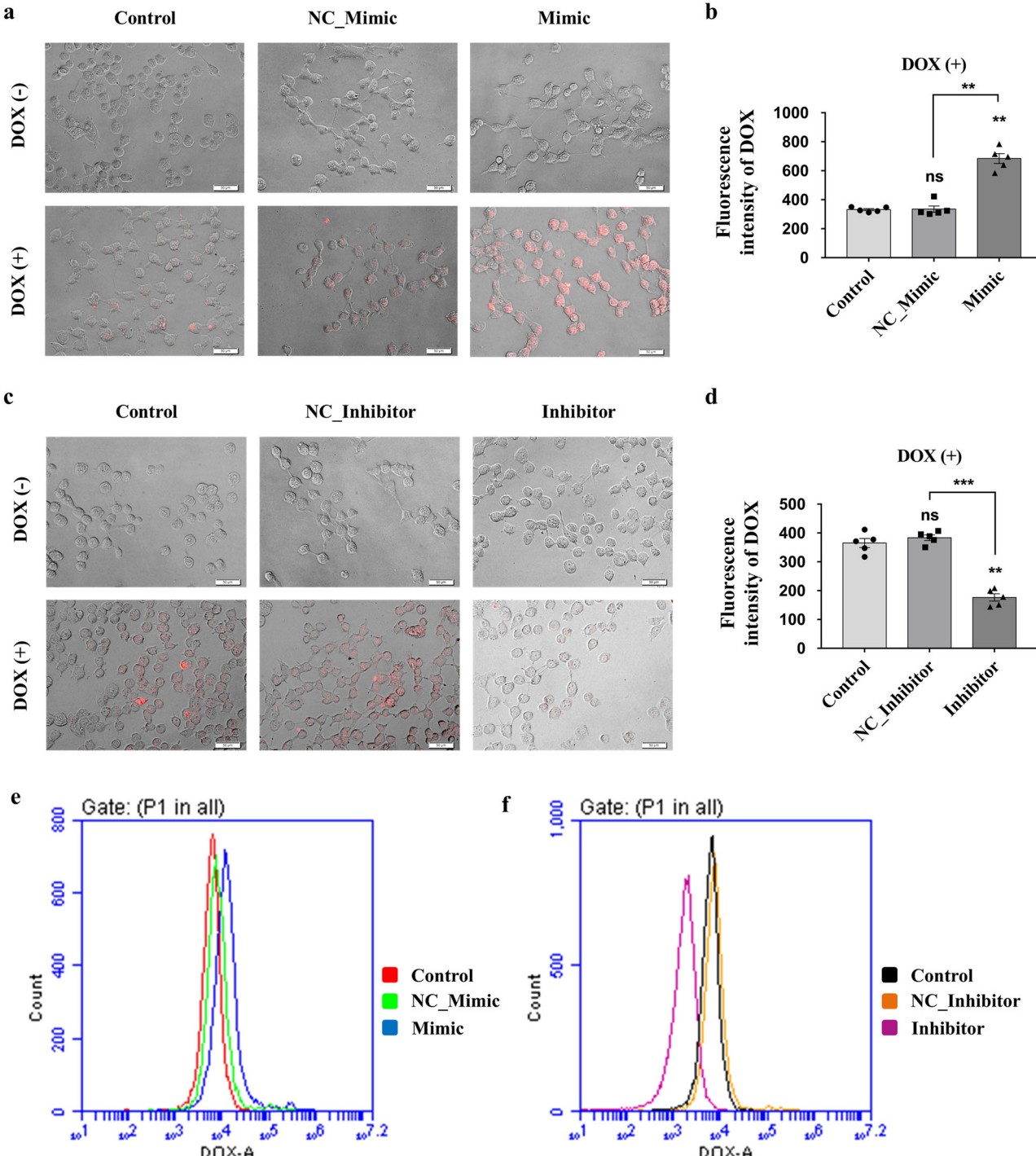

**Fig. 3 Effect of piR-39980 on DOX-accumulation in HT1080 cells. a** Intracellular DOX-accumulation was evaluated by fluorescence microscopy. Scale bar: 50 μm. **b** Intracellular DOX-accumulation was quantified by multimode microplate reader upon transfecting the cells with 20 nM piR-39980 mimic/ NC_Mimic and treating with 0.4 μM DOX. Bars, mean ± SEM; $n = 5$ independent experiments; ns nonsignificant, **$P < 0.01$, Tukey's multiple comparisons test. **c** Intracellular DOX-accumulation was evaluated by fluorescence microscopy. Scale bar: 50 μm. **d** Intracellular DOX-accumulation was quantified by multimode microplate reader upon transfecting the cells with 20 nM piR-39980 inhibitor/ NC_Inhibitor and treating with 0.4 μM DOX. Bars, mean ± SEM; $n = 5$ independent experiments; ns nonsignificant, **$P < 0.01$, ***$P < 0.001$, Tukey's multiple comparisons test. **e, f** DOX-accumulation was measured by flow cytometry transfected with piR-39980 mimic and inhibitor, respectively.

transfection with mimic (20 nM) in combination with DOX (0.4 μM) ($P < 0.05$, Fig. 6b). From these results, we concluded that combination treatment of piR-39980 mimic and DOX increase DNA fragmentation and γH2AX accumulation, suggesting a promising synergistic effect of piR-39980 and DOX on DNA damage-induced apoptosis.

**_RRM2_ and _CYP1A2_ are direct targets of piR-39980**. To decrypt underlying molecular mechanisms by which piR-39980 modulates chemosensitivity of HT1080 cells, we predicted its target genes that relate to chemoresistance and cancer progression. Due to the lack of clinical gene expression data of DOX resistance fibrosarcoma, we extracted potential genes by data mining from

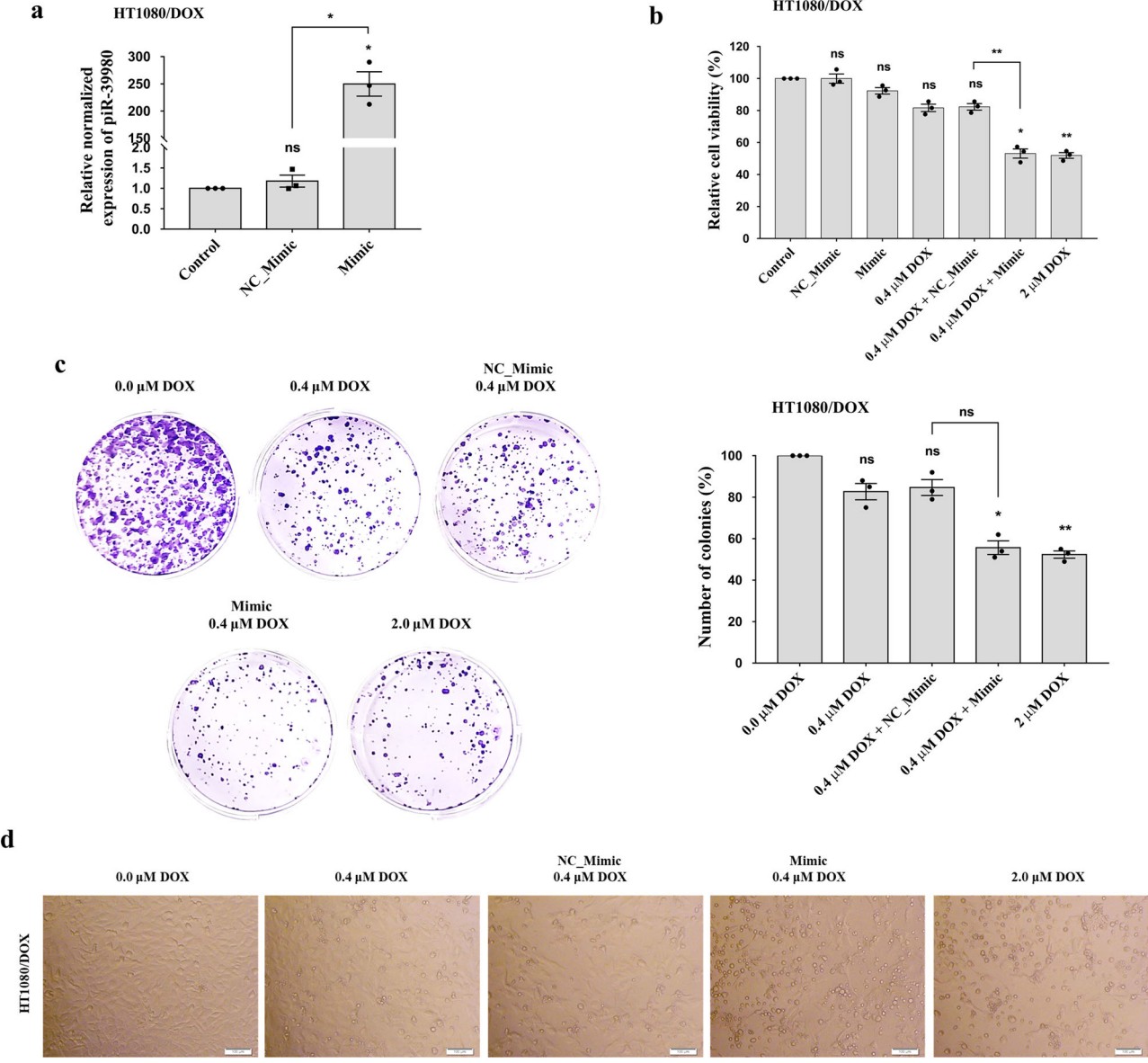

**Fig. 4 Effect of piR-39980 on DOX sensitivity of the resistant HT1080/DOX cells. a** Transfection efficiency showing relative expression of piR-39980 increased by ~250-fold in HT1080/DOX cells transfected with 20 nM piR-39980 mimics compared to controls. Bars, mean ± SEM; $n = 3$ independent experiments; ns nonsignificant, *$P < 0.05$, Tukey's multiple comparisons test. **b** Effect of piR-39980 on HT1080/DOX cell viability determined by MTT assay. Bars, mean ± SEM; $n = 3$ independent experiments; ns nonsignificant, *$P < 0.05$, **$P < 0.01$, Tukey's multiple comparisons test. **c** The colony-forming ability of HT1080/DOX cells transfected with piR-39980 mimic. Bars, mean ± SEM; $n = 3$ independent experiments; ns non-ignificant, *$P < 0.05$, **$P < 0.01$, Tukey's multiple comparisons test. **d** Effect of piR-39980 on HT1080/DOX cell morphology upon 0.4 μM DOX treatment. Scale bar: 100 μm.

previous studies. We obtained 14 genes (*CYP1A1, CYP1A2, ABCB1, GSTP1, MVP, EPHX1, RRM1, RRM2, ABCC3, ABCC6, JUNB, CLU, TOP2A*, and *MCM4*) that are reported to be involved in DOX resistance in different cancers (refer to Method section), but not in sarcoma. We then used these genes to predict targets of piR-39980 using miRanda and found two target genes, *RRM2* and *CYP1A2*. We found piR-39980 targets at 2756-2784 regions of 3′ UTR of *RRM2* (alignment score: 223 and binding energy: −59.619 kCal/mol) with a mismatch at 15th positions in the secondary seed region (Supplementary Fig. 5). We also found piR-39980 targets at 2453–2481 regions of 3′ UTR of *CYP1A2* (alignment score: 231 and binding energy: −66.349 kCal/mol) with a wobble pairing at 15th position within the secondary seed site (Supplementary Fig. 6).

RRM2 is the catalytic subunit of Ribonucleotide reductase (RR), which synthesizes deoxyribonucleotides from ribonucleotides. Thus, RRM2 can modulate cancer cell proliferation, DNA repair, and apoptosis by regulating replication and transcription[34]. RRM2 is also involved in DOX resistance in many cancers[35–37]. We checked the expression of *RRM2*, which was found to be significantly higher in HT1080/DOX cells compared to HT1080 cells ($P = 0.0349$, Fig. 7a). We then transfected piR-39980 mimic and inhibitor individually into the HT1080/DOX cells and HT1080 cells, respectively, to investigate whether piR-39980 can modulate *RRM2* expression. We observed *RRM2* expression was ~2.5-fold decreased upon mimic transfection and ~3.5-fold increased upon inhibitor transfection compared to the respective negative control in HT1080/DOX cells

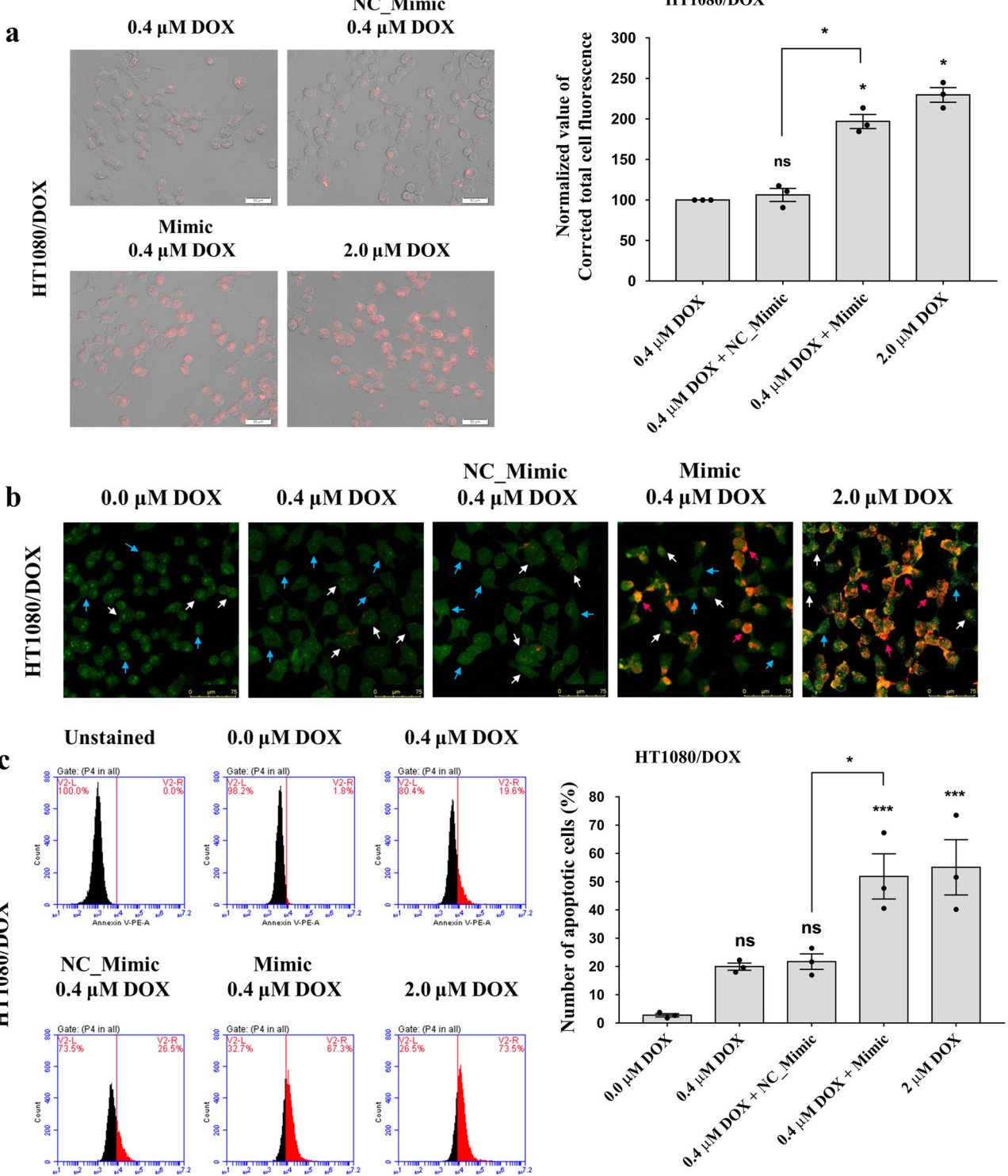

**Fig. 5 Effect of piR-39980 on DOX accumulation and apoptosis of the resistant HT1080/DOX cells. a** Intracellular DOX-accumulation in HT1080/DOX cells in different conditions. Scale bar: 50 μm. Bars, mean ± SEM; $n = 3$ independent experiments; ns nonsignificant, *$P < 0.05$, Tukey's multiple comparisons test. **b** AO/EB dual staining assay showing an increase in apoptotic cell death. Blue, white, and pink arrows indicate live cells, early apoptotic cells, and late apoptotic cells, respectively. Scale bar: 75 μm. **c** Flow cytometric PE Annexin-V apoptosis assay showing an increase in apoptotic cell death upon transfection with piR-39980 mimic and DOX. Bars, mean ± SEM; $n = 3$ independent experiments; ns nonsignificant, *$P < 0.05$, ***$P < 0.001$, Tukey's multiple comparisons test.

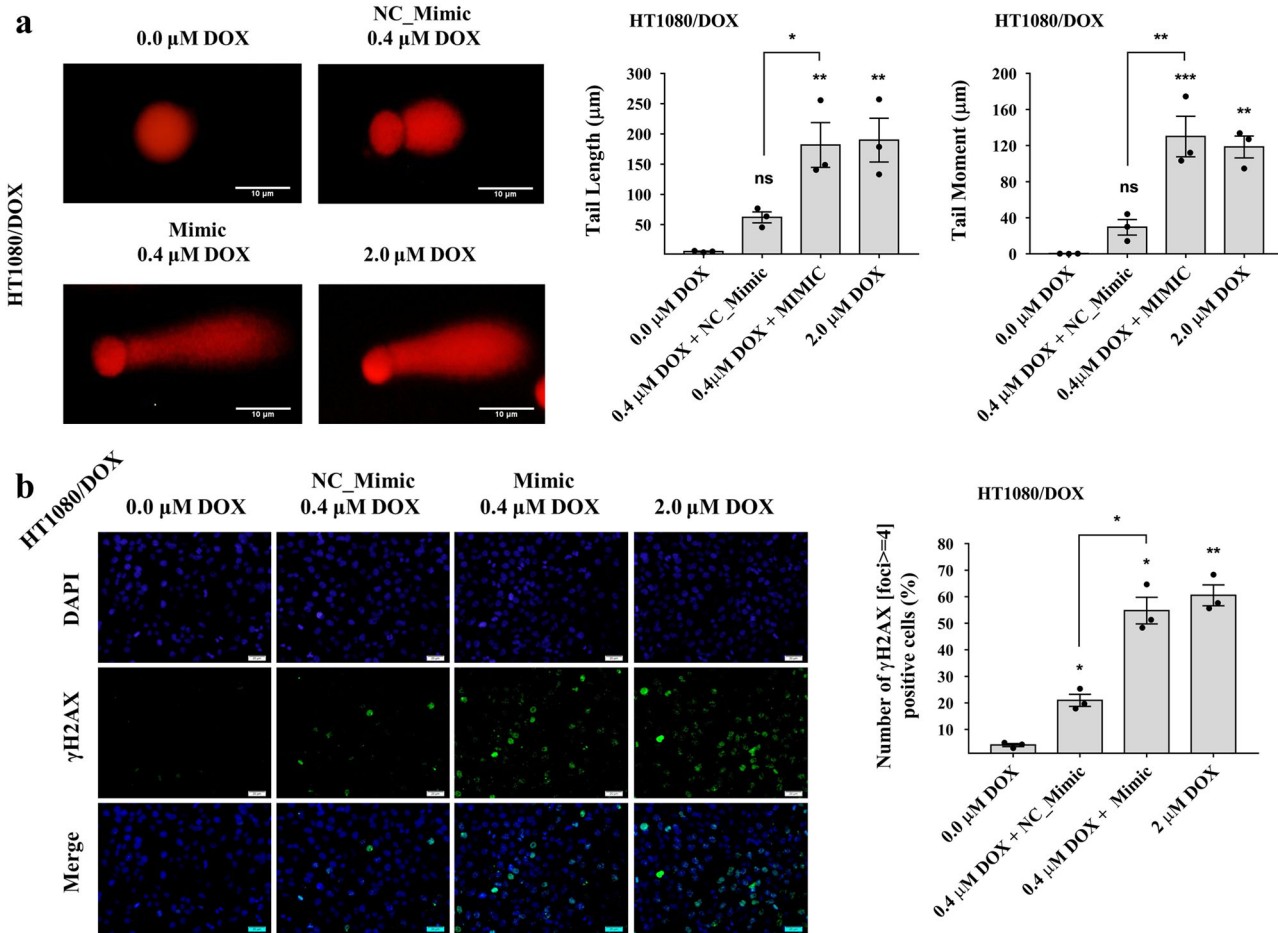

**Fig. 6 Effect of piR-39980 on DOX-induced DNA damage of HT1080/DOX cells. a** Effect of piR-39980 on DOX-induced DNA damage determined by comet assay. Scale bar: 10 µm. Bars, mean ± SEM; $n = 3$ independent experiments; ns nonsignificant, *$P < 0.05$, **$P < 0.01$, ***$P < 0.001$, Tukey's multiple comparisons test. **b** Assay of γH2AX accumulation, a marker of DNA damage, detected by fluorescence imaging using anti- γH2AX antibody. Green fluorescence, DyLight 488 conjugated secondary antibody showing accumulation of γH2AX. Blue fluorescence of DAPI showing nucleus. Scale bar: 20 µm. Bars, mean ± SEM; $n = 3$ independent experiments; ns nonsignificant, *$P < 0.05$, **$P < 0.01$, Tukey's multiple comparisons test.

($P < 0.05$, Fig. 7b). Whereas, in HT1080 cells, *RRM2* expression was ~2.0-fold decreased after mimic transfection and ~2.5-fold increased after piRNA inhibitor transfection compared to the respective negative control ($P < 0.05$, Fig. 7c).

It was reported that CYP1A2, a phase I/II metabolizing enzyme, was upregulated in DOX-resistant breast cancer cells. Upregulation of this enzyme regulates the drug resistance mechanism by deactivating DOX[13]. Volkova et al.[38] reported that CYP1A2 expression was increased in cardiac cells when exposed to DOX. This upregulation of CYP1A2 provided a protective response against DOX-induced cardiac toxicity[38]. We checked the expression of *CYP1A2* in HT1080/DOX cells and found this gene to be significantly highly expressed compared to HT1080 cells ($P = 0.0124$, Fig. 7d). We then transfected piR-39980 mimic and inhibitor into the HT1080/DOX cells and HT1080 cells. In HT1080/DOX cells, we observed ~3.5-fold drop and ~3.5-fold rise in *CYP1A2* expression upon transfection with piRNA mimic and inhibitor respectively compared to the respective negative control ($P < 0.05$, Fig. 7e). Similarly, *CYP1A2* expression was ~2.5-fold decreased upon piRNA mimic transfection and ~3.0-fold increased upon inhibitor transfection compared to the respective negative control in HT1080 cells ($P < 0.01$, Fig. 7f). We conjecture that *RRM2* and *CYP1A2* are possibly targeted by piR-39980, as they show reciprocal expression and were further validated by performing luciferase reporter assay, described in the next section.

Our previous study reported *RRM2* as a direct target of piR-39980 by performing dual-luciferase reporter assay[29]. Again, we performed luciferase assay for *RRM2* along with *CYP1A2* to revalidate. We designed two different reporter assay constructs for each gene by cloning wild-type and mutant (mutated fourth to eighth bases at primary seed site) 3'-UTR of *RRM2/CYP1A2* into psiCHECK-2 vectors (Supplementary Figs. 7, 8). We performed luciferase assay by co-transfecting reporter assay constructs (WT/MUT-RRM2/CYP1A2) with piR-39980 mimic/ NC_Mimic in HEK293 human embryonic kidney cell line, using Lipofectamine-2000 reagent, and then measured relative luciferase activity [RLU (Firefly)/RLU (Renilla)] after 24 h. We observed a significant decrease in luciferase activity by 50% for wild-type *RRM2* ($P < 0.001$, Fig. 7g), and by 65% for wild-type *CYP1A2* ($P < 0.001$, Fig. 7h) 3'-UTR co-transfected with mimic compared to NC_Mimic. However, we did not find any significant reduction in luciferase activity of mutant *RRM2/CYP1A2* 3'-UTR constructs co-transfected with piR-39980 mimic. These results confirmed that piR-39980 reduces *RRM2* and *CYP1A2* expression by directly targeting their 3'-UTR, which in turn might be inducing DOX sensitivity.

**piR-39980/RRM2 axis modulates DOX-induced cell death in fibrosarcoma**. We performed a series of functional assays to

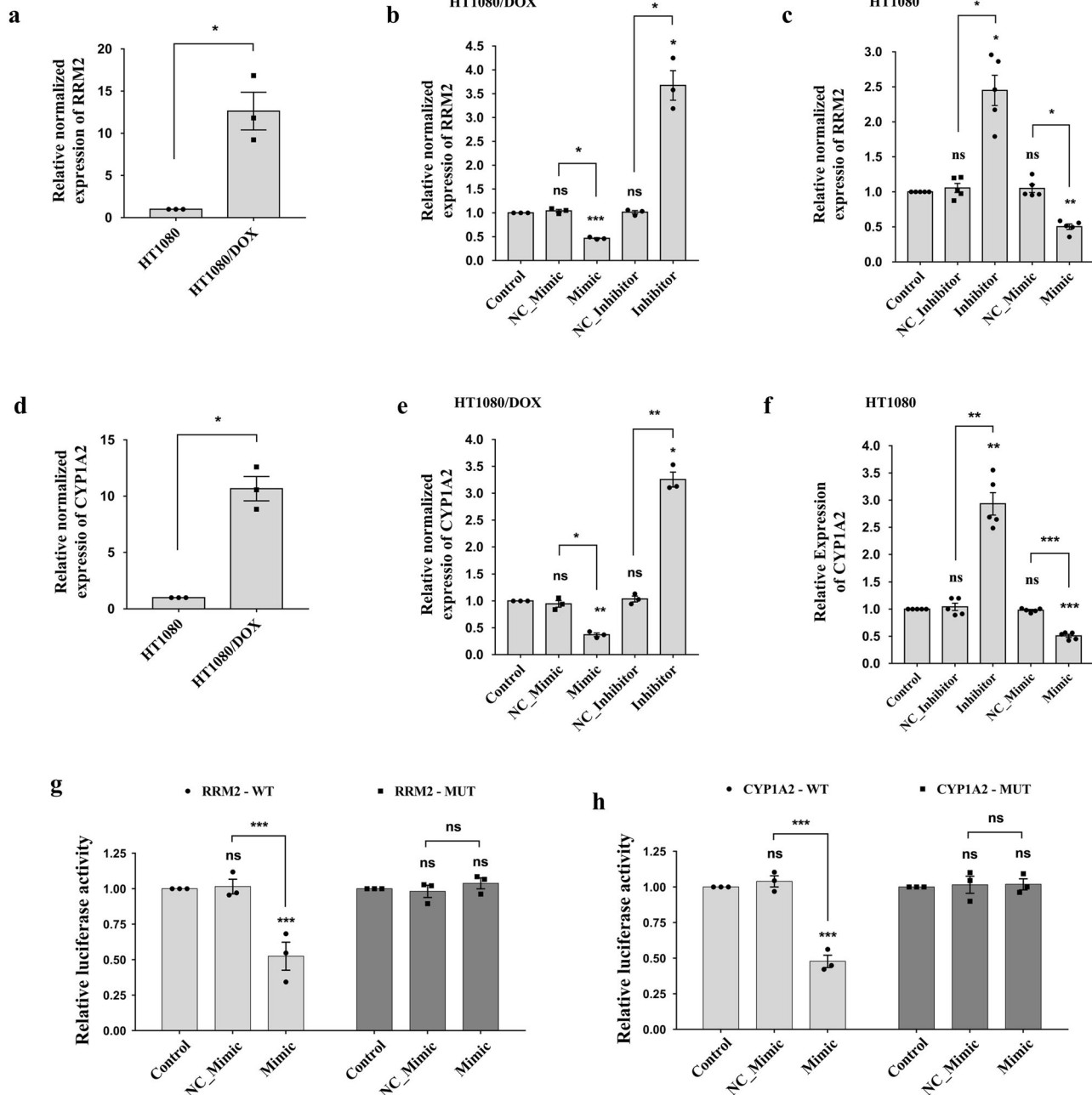

**Fig. 7 RRM2 and CYP1A2 are the direct functional targets of piR-39980. a** Relative expression of *RRM2* detected in parental HT1080 and resistant HT1080/DOX cell lines by qRT-PCR. Bars, mean ± SEM; *n* = 3 independent experiments; *P < 0.05, *t*-test. **b** The expression of *RRM2* in HT1080/DOX cells upon transfected with 20 nM piR-39980 mimic/inhibitor as detected by qRT-PCR. Bars, mean ± SEM; *n* = 3 independent experiments; ns nonsignificant, *P < 0.05, ***P < 0.001, Tukey's multiple comparisons test. **c** The expression of *RRM2* in HT1080 cells upon transfected with 20 nM piR-39980 inhibitor/ mimic as detected by qRT-PCR. Bars, mean ± SEM; *n* = 5 independent experiments; ns nonsignificant, *P < 0.05, **P < 0.01, Tukey's multiple comparisons test. **d** Relative expression of *CYP1A2* detected in parental HT1080 and resistant HT1080/DOX cell lines by qRT-PCR. Bars, mean ± SEM; *n* = 3 independent experiments; *P < 0.05, *t*-test. **e** The expression of *CYP1A2* in HT1080/DOX cells upon transfected with 20 nM piR-39980 mimic/inhibitor compared to control, detected by qRT-PCR. Bars, mean ± SEM; *n* = 3 independent experiments; ns nonsignificant, *P < 0.05, **P < 0.01, Tukey's multiple comparisons test. **f** The expression of *CYP1A2* in HT1080 cells transfected with 20 nM piR-39980 mimic/inhibitors detected by qRT-PCR. Bars, mean ± SEM; *n* = 5 independent experiments; ns nonsignificant, **P < 0.01, ***P < 0.001, Tukey's multiple comparisons test. **g** Luciferase reporter constructs containing wild-type/mutant piR-39980 target site in *RRM2* 3'-UTR were co-transfected with mimic into the HEK293 cells. Bars, mean ± SEM; *n* = 3 independent experiments; ns nonsignificant, ***P < 0.001, Sidak's multiple comparisons test. **h** Luciferase reporter constructs containing wild-type/mutant piR-39980 target site in *CYP1A2* 3'-UTR were co-transfected with mimic into the HEK293 cells. Bars, mean ± SEM; *n* = 3 independent experiments; ns nonsignificant, ***P < 0.001, Sidak's multiple comparisons test.

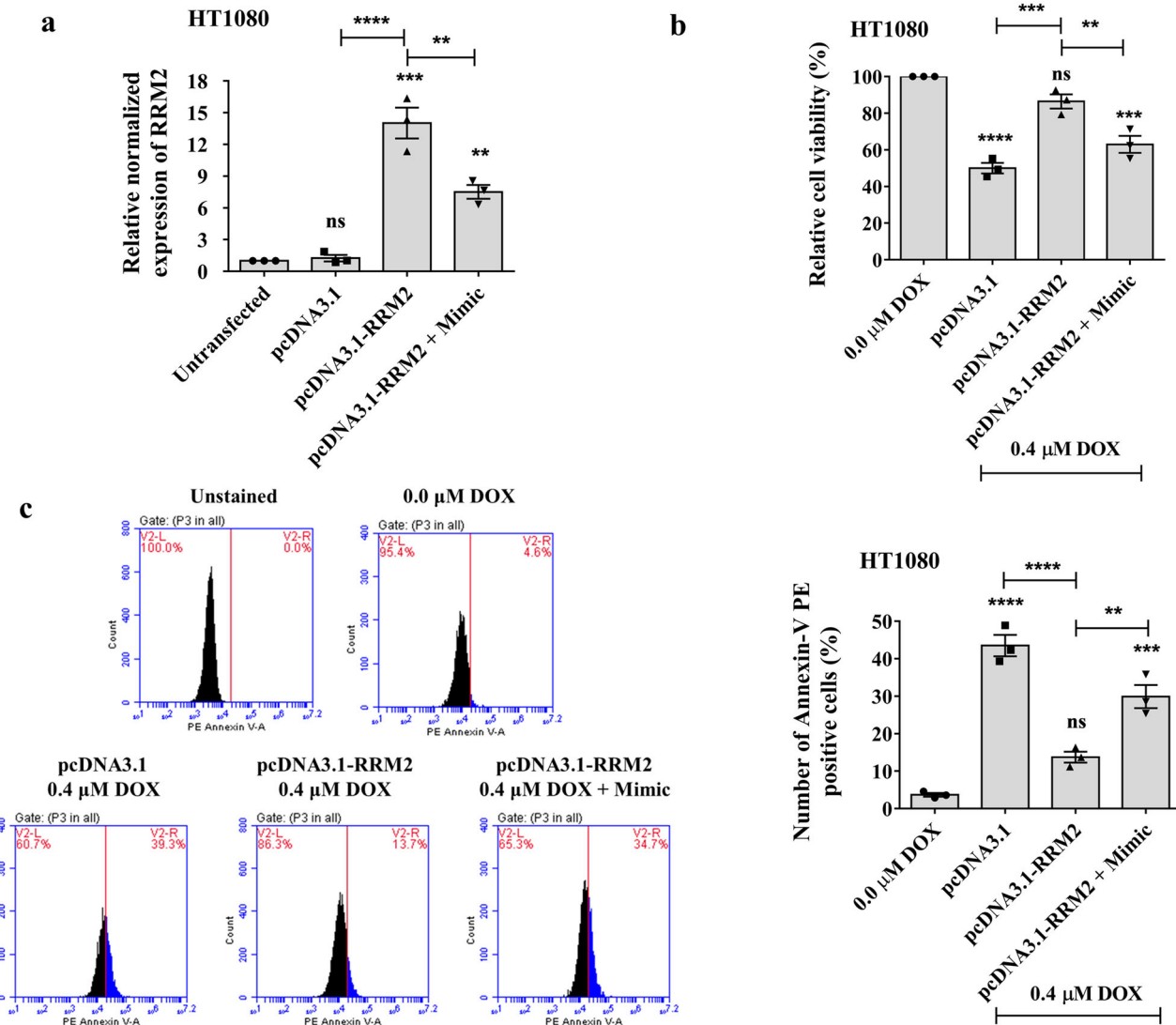

**Fig. 8 piR-39980/RRM2 regulates cell viability and apoptosis of HT1080 cells upon DOX treatment. a** Relative expression of *RRM2* was detected by qRT-PCR in HT1080/DOX cells transfected with pcDNA3.1_RRM2 compared with pcDNA3.1. Bars, mean ± SEM; $n = 3$ independent experiments; ns nonsignificant, **$P < 0.01$, ***$P < 0.001$, ****$P < 0.0001$, Tukey's multiple comparisons test. **b** MTT assay showing role of RRM2/piR-39980 axis in HT1080 cell viability. Bars, mean ± SEM; $n = 3$ independent experiments; ns nonsignificant, **$P < 0.01$, ***$P < 0.001$, ****$P < 0.0001$, Tukey's multiple comparisons test. **c** PE Annexin-V flow cytometry assay showing role of RRM2/piR-39980 axis on apoptosis of HT1080 cells and quantification of apoptotic cells upon transfection with pcDNA3.1_RRM2 and piR-39980. Bars, mean ± SEM; $n = 3$ independent experiments; ns nonsignificant, **$P < 0.01$, ***$P < 0.001$, ****$P < 0.0001$, Tukey's multiple comparisons test.

decipher the mechanism through which piR-39980 and its targets modulate the chemoresistance of HT1080 cells. We designed the RRM2 overexpression construct (pcDNA3.1_RRM2) by cloning CDS of RRM2 with partial 3′-UTR sequence containing piR-39980 binding site into pcDNA3.1 vector. Empty pcDNA3.1 vector was used as control. For functional assays, HT1080 cells were transfected with pcDNA3.1, pcDNA3.1_RRM2, and co-transfected with pcDNA3.1_RRM2 and piR-39980 mimic. The cells were treated with 0.4 μM DOX 24 h after transfection to conduct functional assays.

At first, we checked the expression of *RRM2* in HT1080 cells upon transfection of pcDNA3.1_RRM2 and co-transfection of pcDNA3.1_RRM2 with piR-39980 mimic (Fig. 8a). HT1080 cells transfected with pcDNA3.1_RRM2 showed a significant upregulation of *RRM2* mRNA level compared to cells transfected with pcDNA3.1 ($P < 0.001$). However, the expression of *RRM2* was partially restored by piR-39980 in the cells co-transfected with pcDNA3.1_RRM2 and piR-39980 mimic ($P < 0.01$). Further, to

investigate the role of piR-39980/RRM2 in modulating chemoresistance in fibrosarcoma, we performed an MTT assay and found that cell viability of DOX-treated HT1080 cells was increased by 35% upon RRM2 overexpression (pcDNA3.1_RRM2 + DOX-treated cells) with respect to the control (pcDNA3.1 + DOX-treated cells) ($P < 0.001$, Fig. 8b). However, cell viability was significantly reduced ($P < 0.01$) in the co-transfected group (pcDNA3.1_RRM2 + Mimic + DOX-treated cells) than the over-expression group (pcDNA3.1_RRM2 + DOX-treated cells). These results suggested that piR-39980 restrains RRM2 mediated induction of cell viability after DOX treatment.

Further, to check whether piR-39980 induces cell death, we detected apoptotic cells by PE Annexin-V assay using flow cytometry (Fig. 8c). We found 30% reduction in Annexin-V positive cells (apoptotic cells) upon overexpression of RRM2 than control (transfected with pcDNA3.1) ($P < 0.0001$, Fig. 8c). On the contrary, the number of apoptotic cells was significantly increased when piR-39980 was co-transfected with pcDNA3.1_RRM2

compared to pcDNA3.1_RRM2 transfectants ($P < 0.01$, Fig. 8c). These results indicate that piR-39980 enhances DOX-induced cell death by repressing RRM2.

### piR-39980/RRM2 modulates DOX-induced DNA damage in fibrosarcoma.

Earlier studies have reported reduced DNA damage and enhanced DNA repair mechanisms confer resistance to anticancer drugs[39]. RRM2 facilitates DNA repair by catalyzing de novo synthesis of deoxyribonucleotides[36]. Therefore, we investigated the role of piR-39980/RRM2 axis on DOX-induced DNA damage by comet assay (Fig. 9a, b, c and Supplementary Fig. 9). Compared to pcDNA3.1 transfectant, we found a significantly shorter tail in pcDNA3.1_RRM2 transfected HT1080 cells upon DOX treatment ($P < 0.0001$, Fig. 9b), which indicated RRM2 induces DNA repair in DOX-treated cells. However, HT1080 cells showed a significantly large tail when co-transfected by piR-39980 mimic with pcDNA3.1_RRM2 compared to pcDNA3.1_RRM2 transfectants ($P < 0.01$, Fig. 9b). Similarly, the comet tail moment was decreased in HT1080 cells transfected with pcDNA3.1_RRM2 ($P < 0.0001$), whereas piR-39980 significantly restrained tail moment when co-transfected with pcDNA3.1_RRM2 ($P < 0.05$, Fig. 9c). These results suggested that

piR-39980 induces DOX-mediated DNA damage by inhibiting RRM2 expression.

These findings were further confirmed by $\gamma$H2AX accumulation assay (Fig. 9d). Interestingly, $\gamma$H2AX foci (the number of foci >4) containing cells were greatly reduced (~3.5-fold) in DOX-treated cells due to RRM2 overexpression (Fig. 9d). However, $\gamma$H2AX foci containing cells were increased upon co-transfection of piR-39980 with pcDNA3.1_RRM2, which indicates piR-39980 restrains RRM2 mediated DNA repair (Fig. 9d). These results confirmed piR-39980 promotes DOX-induced DNA damage and inhibits DNA repair by repressing RRM2.

These results confirmed that piR-39980 is the negative regulator of *RRM2*, and the silencing of *RRM2* by piRNA inhibits DNA repair, facilitating DOX-induced DNA damage; therefore, cells pass into apoptotic cell death. Taken together, piR-39980 increases the DOX sensitivity of fibrosarcoma cells by repressing *RRM2*.

### piR-39980/CYP1A2 modulates DOX-accumulation and apoptosis in fibrosarcoma.

Like reduced DNA damage and enhanced DNA repair mechanisms, failure of DOX accumulation due to metabolism by cancer cells is one of the critical regulators of DOX

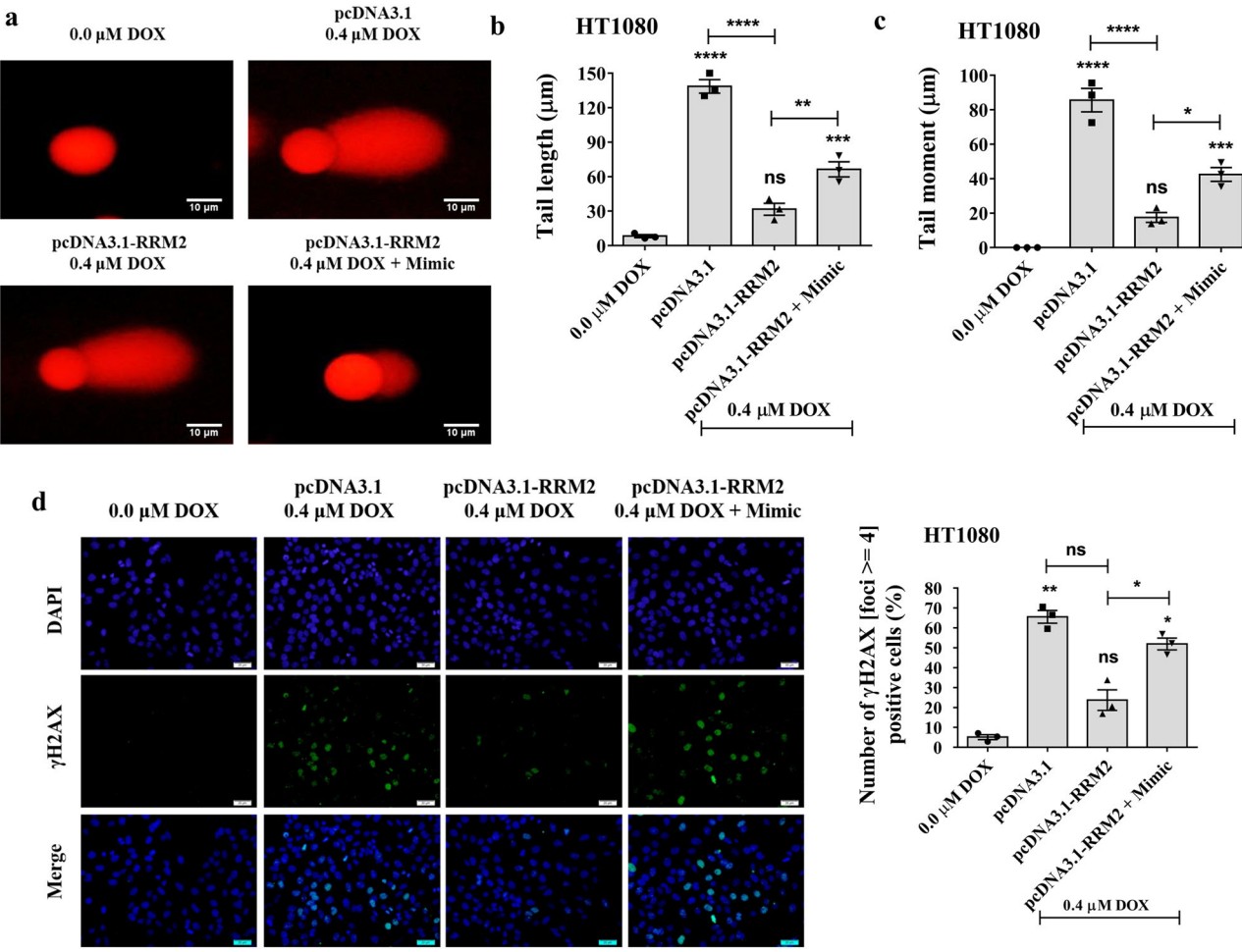

**Fig. 9 piR-39980/RRM2 regulates $\gamma$H2AX accumulation and apoptosis of HT1080 cells upon DOX treatment. a** Comet assay showing the impact of RRM2/piR-39980 axis on DNA damage of HT1080 cells. Scale bar: 10 $\mu$m. **b** Comet tail length and **c** comet tail moment measured by ImageJ. Bars, mean ± SEM; $n = 3$ independent experiments; ns nonsignificant, *$P < 0.05$, **$P < 0.01$, ***$P < 0.001$, ****$P < 0.0001$, Tukey's multiple comparisons test. **d** $\gamma$H2AX accumulation assay detected by fluorescence imaging using anti-$\gamma$H2AX antibody and quantification of $\gamma$H2AX- foci upon transfection with pcDNA3.1_RRM2 and piR-39980. Green fluorescence, DyLight 488 conjugated secondary antibody showing accumulation of $\gamma$H2AX. Blue fluorescence of DAPI showing nucleus. Scale bar: 20 $\mu$m. Bars, mean ± SEM; $n = 3$ independent experiments; ns nonsignificant, *$P < 0.05$, **$P < 0.01$, Tukey's multiple comparisons test.

resistance that cause the failure of treatments[40]. We accomplished a series of functional assays to inspect the role of piR-39980/CYP1A2 in modulating the chemoresistance of HT1080 cells through regulating DOX accumulation. We designed the CYP1A2 overexpression construct (pcDNA3.1_CYP1A2) by cloning CDS of CYP1A2 with a partial 3′-UTR sequence containing piR-39980 binding site into pcDNA3.1 vector. For all functional assays, HT1080 cells were transfected with pcDNA3.1, pcDNA3.1_CYP1A2, and co-transfected with pcDNA3.1_CYP1A2 and piR-39980 mimic followed by treatment with 0.4 μM DOX after 24 h of transfection.

At first, we checked the expression of *CYP1A2* in HT1080 cells upon transfection of pcDNA3.1_CYP1A2 and co-transfection of pcDNA3.1_CYP1A2 with piR-39980 mimic (Fig. 10a). HT1080 cells transfected with pcDNA3.1_CYP1A2 showed a significant upregulation of *CYP1A2* mRNA level compared to cells transfected with pcDNA3.1 ($P < 0.001$). However, cells co-transfected with pcDNA3.1_CYP1A2 and piR-39980 mimic showed partial restoration of *CYP1A2* expression by piR-39980 ($P < 0.01$). To investigate the role of piR-39980/CYP1A2 in modulating chemoresistance in fibrosarcoma, we performed an MTT assay and found that the cell viability of the DOX-treated

cell was increased by 40% upon CYP1A2 overexpression ($P < 0.05$, Fig. 10b). However, piR-39980 restrains CYP1A2 mediated induction of cell viability after DOX treatment.

Then, we investigated the role of the piR-39980/CYP1A2 axis on DOX accumulation by fluorescent cell imaging and quantification of intracellular DOX (Fig. 10c). We found significant reduction in intracellular DOX in pcDNA3.1_CYP1A2 transfected HT1080 cells compared to pcDNA3.1 transfectants ($P < 0.05$, Fig. 10d). However, intracellular DOX was increased by 30% when piR-39980 mimic was co-transfected with pcDNA3.1_CYP1A2 compared to pcDNA3.1_CYP1A2 alone ($P < 0.01$, Fig. 10d). The result indicated that piR-39980 increased DOX accumulation by targeting *CYP1A2*. We also performed an apoptosis assay using flow cytometry (Fig. 10e) to confirm the possibility that piR-39980 mediated DOX accumulation by targeting *CYP1A2* induces cell death. We found 25% reduction in apoptotic cells upon overexpression of CYP1A2 ($P < 0.01$, Fig. 10f). However, the number of apoptotic cells was increased by 15% when piR-39980 was co-transfected with pcDNA3.1_CYP1A2 compared to pcDNA3.1_CYP1A2 transfectants (Fig. 10f).

These results suggested that piR-39980 is the negative regulator of *CYP1A2*, which mediates the silencing of *CYP1A2* that induces

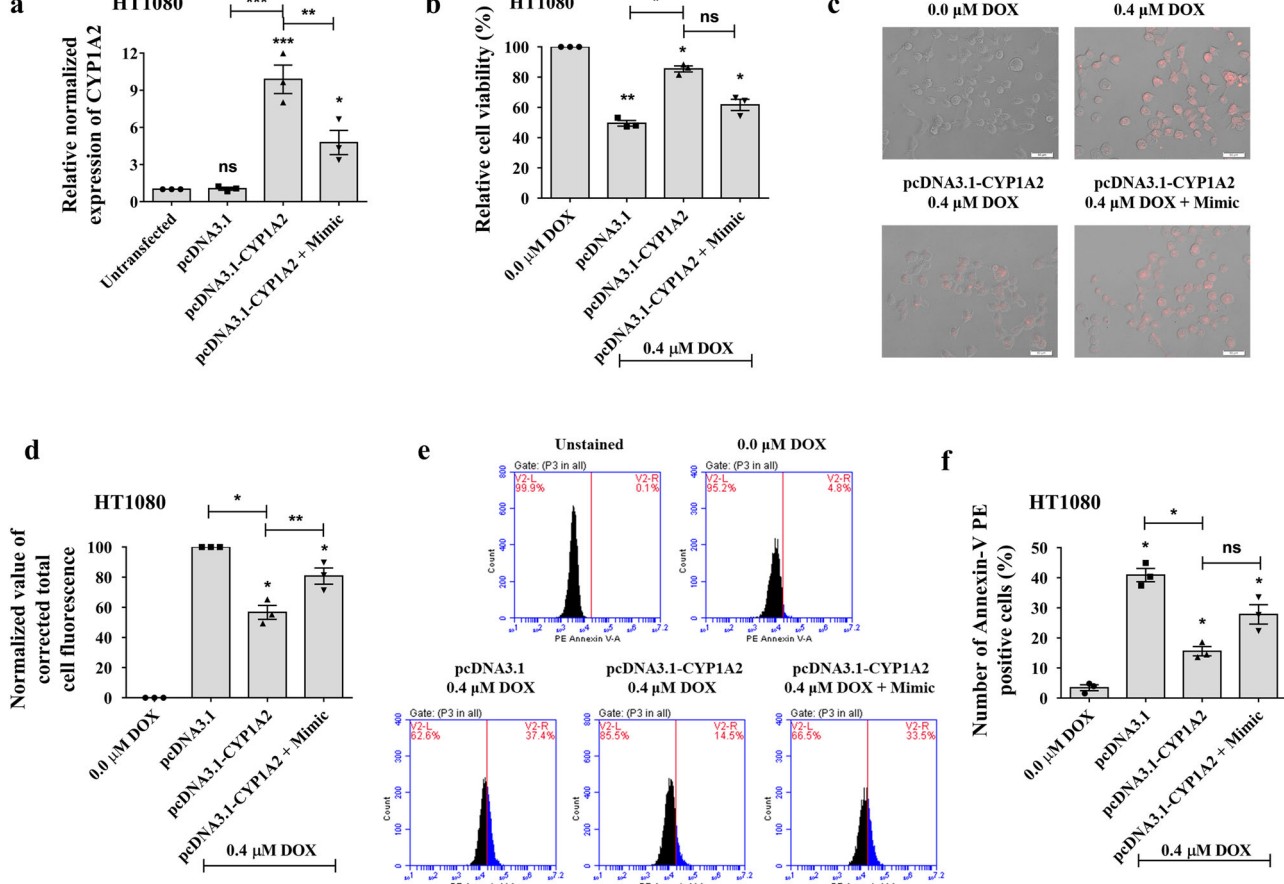

**Fig. 10 piR-39980/CYP1A2 regulates DOX-accumulation and apoptosis in HT1080 cells. a** Relative expression of *CYP1A2* was detected by qRT-PCR in HT1080/DOX cells transfected with pcDNA3.1_CYP1A2 and empty pcDNA3.1 vector. Bars, mean ± SEM; $n = 3$ independent experiments; ns nonsignificant, \*$P < 0.05$, \*\*$P < 0.01$, \*\*\*$P < 0.001$, Tukey's multiple comparisons test. **b** Role of CYP1A2/piR-39980 axis in HT1080 cell viability determined by MTT assay. Bars, mean ± SEM; $n = 3$ independent experiments; ns nonsignificant, \*$P < 0.05$, \*\*$P < 0.01$, Tukey's multiple comparisons test. **c** Role of CYP1A2/piR-39980 axis in intracellular DOX accumulation determined by fluorescence microscopy. Scale bar: 50 μm. **d** The intensity of intracellular DOX was quantified by ImageJ using the images of Fig. 10c. Bars, mean ± SEM; $n = 3$ independent experiments; \*$P < 0.05$, \*\*$P < 0.01$, Tukey's multiple comparisons test. **e** PE Annexin-V flow cytometry assay showing the role of CYP1A2/piR-39980 axis in HT1080 cell death. **f** Quantification of apoptotic cells upon transfection with pcDNA3.1_CYP1A2 and piR-39980. Bars, mean ± SEM; $n = 3$ independent experiments; ns nonsignificant, \*$P < 0.05$, Tukey's multiple comparisons test.

intracellular DOX accumulation. Finally, an increased level of DOX within the cells due to enhanced intracellular DOX accumulation causes cell death. Taken together, piR-39980 increases DOX sensitivity of fibrosarcoma cells by repressing CYP1A2 in addition to RRM2.

## Discussion

Although the use of chemotherapeutic agents has dramatically improved patients' survival over the last decade, the development of chemoresistance remains the major obstacle to chemotherapy for the successful treatment of cancer[16]. Statistical data have demonstrated that about 90% of death of cancer patients is due to chemoresistance[41]. Chemoresistance may be intrinsic or acquired that is developed during chemotherapy. Acquired drug resistance is a multifactorial phenomenon. Numerous studies have shown that cancer cells can develop resistance against any effective anticancer drug by diverse mechanisms. These mechanisms include reduced drug uptake, augmented drug efflux, increased drug metabolism, over-activation of DNA repair cascade, decreased DNA damage, evasion of drug-induced apoptosis, etc. Activation of any one or more of these mechanisms restrains the efficacy of chemotherapeutic drugs, which poses difficulty in treatment and ultimately results in poor therapeutic outcome[42].

Bao et al.[43] has demonstrated that increased P-glycoprotein expression is associated with DOX resistance in metastatic breast cancer cells[43]. Pisco et al.[44] have shown reduced accumulation of DOX in drug-resistant leukemia cells is due to decreased DOX-uptake[44]. In addition, DOX-resistant breast adenocarcinoma (MCF7/DOX) cells exhibit high DNA repair potential and hence are less susceptible to radiation-induced DNA damage and apoptosis[45]. Recently, Wang et al.[40] profiled different DOX metabolites in the wild-type and drug-resistant cells, which showed DOX metabolites are different from those in the liver or kidney. This finding indicates tumor cells and drug-resistant tumor cells harbor unique DOX-metabolism pathways. This DOX metabolism leads to decrease intracellular DOX that fails to induce cell death, causing chemoresistance[40].

Over the last decades, research is mainly focused on genetic and epigenetic factors that induce chemoresistance by altering drug resistance pathways. However, recent evidence suggests that dysregulation of ncRNAs plays a key regulatory role in chemoresistance[46], especially miRNAs. The role of miRNAs in the modulation of chemoresistance is studied in several cancers, including breast cancer, ovarian cancer, prostate cancer, lung cancer, colon cancer, and leukemia[47]. These studies showed overexpression or silencing of miRNAs could effectively overcome drug resistance in cancer therapy. For instance, role of miRNAs such as miR-143, miR-124, miR-382, miR-708, miR-34c, miR-199a-3p, miR-29b-1, miR-30a, and miR-101 are reported in sarcoma increasing DOX sensitivity[48]. Recently, piRNAs, a subclass of small ncRNAs, emerged as a key player in cancer by modulating proliferation, migration, invasion, apoptosis, and angiogenesis[49]. There are only a handful of reports on piRNAs modulating chemoresistance in cancers, such as colorectal cancer, breast cancer, and neuroblastoma[24,27,28]. However, there are no reports yet on piRNA mediating chemoresistance/chemosensitivity in human fibrosarcoma.

In the present study, we demonstrated that piR-39980 is expressed at a lower level in DOX-resistant fibrosarcoma cell line HT1080/DOX compared to the parental HT1080 cells (Fig. 1e). Overexpression of piR-39980 in HT1080 cells promotes DOX-induced cell apoptosis and anti-proliferative effects by increasing intracellular DOX-accumulation, whereas silencing of piR-39980 decreased DOX-accumulation and enhanced cell viability (Figs. 1f, g, 2, and 3). In particular, piR-39980 mimic transfection

into HT1080/DOX cells induced intracellular DOX-accumulation, DNA damage, and apoptotic cell death (Figs. 4, 5, and 6). These results only revealed piR-39980 increases the sensitivity of DOX-resistant cells to DOX, but not the mechanisms through which piRNA does this function. Hence, we predicted possible targets of piR-39980 that are related to drug metabolism using miRanda and found that piR-39980 targets 3′-UTR of RRM2 and CYP1A2, which was further confirmed by Dual-luciferase reporter assay. RRM2 and CYP1A2 are significantly upregulated in HT1080/DOX cells compared to HT1080 parental cells but were repressed upon transfection with piR-39980 mimic (Fig. 7).

RRM2 is the catalytic subunit of the nucleotide metabolism enzyme, ribonucleotide reductase, crucial for DNA replication and repair by synthesizing dNTPs. RRM2 degradation induces genome instability because of failures in DNA repair due to depletion of the dNTP pool[50]. Accumulating evidence suggests that increased DNA repair processes play a central role in the development of drug resistance. DNA damage-mediated cell death induced by chemotherapeutic drugs is removed by DNA repair mechanisms, which helps cancer cells to survive[51]. The pernicious effect of many chemotherapeutic drugs, including DOX, relies on their ability to damage DNA. DNA damage tolerance influenced by repair offers an alternative route to chemoresistance to a certain drug[52].

Overexpression of RRM2 is associated with the development of resistance to multiple chemotherapeutic drugs such as gemcitabine and imatinib, including DOX. RRM2 was upregulated in DOX-resistant human breast cancer cells and peripheral blood samples of imatinib-resistant leukemia patients[36,53]. Long-term gemcitabine treatment in pancreatic cancer cells induced resistance against the drug, which was reversed by the silencing of RRM2[54]. Interestingly, our study also revealed significant upregulation of RRM2 in DOX-resistant HT1080 cells. Overexpression of RRM2 in HT1080 cells induced DNA repair and reduced apoptotic cell death, which was restored after piR-39980 mimic transfection. Overexpression of piR-39980 in HT1080 cells increased DNA damage and apoptosis by restraining RRM2 expression (Figs. 8 and 9). These results suggest that the repression of RRM2 by piR-39980 contributes to increased DOX sensitivity.

Like reduced DNA damage and enhanced DNA repair mechanisms, failure of drug accumulation in cancer cells is one of the critical regulators of drug resistance, causing treatment failures. Lower drug uptake and enhanced drug efflux, which lower intracellular drug accumulation, are well-studied mechanisms to restrain chemoresistance. Moreover, increased drug metabolism has emerged as a potential mechanism to decrease intracellular drug accumulation that confers chemoresistance[55]. Studies on DOX metabolism have shown that reduced DOX within cells due to drug metabolism significantly affects the treatment outcome of this drug[40]. Remarkably, the metabolites of DOX cause serious side effects. For instance, 7-deoxydoxorubicinone and DOXol, two DOX metabolites, are known to be associated with cardiotoxicity[40,56]. Therefore, inhibition of DOX metabolism within the cancer cells may reduce chemoresistance and DOX metabolites mediated side effects.

Surprisingly, we found significant upregulation of cytochrome P450 enzyme, CYP1A2 in DOX-resistant HT1080 cells in the current study. CYP1A2 is a major drug-metabolizing enzyme that belongs to the CYP1 family of the cytochrome P450 superfamily enzymes[57]. Previously, CYP1A2 was found to be upregulated by 96-fold in DOX-resistant MCF7 breast cancer cells[13]. Increased CYP1A2 activity was associated with an increased risk of breast cancer[57]. We demonstrated that overexpression of piR-39980 in HT1080 cells led to the repression of CYP1A2. Moreover, intercellular DOX-accumulation was reduced upon overexpression of

CYP1A2, which was restored after piR-39980 mimic transfection (Fig. 10). These results suggest that the repression of *CYP1A2* by piR-39980 contributes to increased sensitivity to DOX by promoting DOX-accumulation.

Remarkably, our previous study has shown that piR-39980 is upregulated in osteosarcoma cells and acts as an onco-piR by targeting the 3′UTR of Leukocyte elastase inhibitor (SERPINB1)[58]. However, the present study has revealed the opposite role of this piRNA in DOX-resistant fibrosarcoma cells. piR-39980 plays a tumor suppressor role in this cancer by inducing DOX sensitivity through targeting *RRM2* and *CYP1A2*. The opposite role of the same piRNA in different cancers is also documented earlier. Cheng et al., [59] reported that piR-823 was downregulated in gastric cancer tissues compared to non-cancerous tissues and played a tumor suppressor role. Over-expression of piR-823 suppressed cell growth in vitro and in vivo[59]. In contrast, Yin et al. [25] revealed that piR-823 was significantly upregulated in colorectal cancer and played an oncogenic role. piR-823 promoted colorectal cancer cell proliferation by upregulating phosphorylation and transcriptional activity of *HSF1*[25]. Similarly, piR-651 was upregulated in lung cancer tissues and promoted tumor growth and metastasis[22,60], whereas this piRNA was underexpressed in the serum of classical Hodgkin lymphoma patients. Lower expression of piR-651 was associated with shorter overall survival and disease-free survival[61]. These studies showed that piRNAs play multifaceted regulatory functions and are cancer-specific. This upraised the possibilities of involvement of some additional factors other than sequence complementarity alone which affect piRNA targeting like that of miRNAs[62], which need to be investigated in the future.

In conclusion, the present study revealed a significant correlation between piR-39980 expression and the response of fibrosarcoma to DOX. More specifically, we demonstrated that RRM2 and CYP1A2 play a vital role in the resistance of human fibrosarcoma to DOX. CYP1A2 confers DOX resistance by inactivating intracellular DOX, whereas RRM2 promotes DOX resistance by inducing a repair mechanism that rescues DOX-mediated DNA damage. In other words, DOX inactivation by CYP1A2 and induction of DNA repair by RRM2 inhibits DOX-mediated cell death. Our finding showed that piR-39980 could attenuate the DOX resistance by repressing the expression of *RRM2* and *CYP1A2* in DOX-resistant fibrosarcoma cells (Supplementary Fig. 10) and hence could be a potential therapeutic agent for improving the clinical response of fibrosarcoma patients to DOX. However, more studies using clinical fibrosarcoma patient samples and in vivo animal models are needed to uphold the association of piR-39980 and its two targets, *RRM2* and *CYP1A2*, with positive responses of fibrosarcoma patients to DOX.

## Methods

**Cell lines and culture conditions**. Human fibrosarcoma cell line (HT1080) and human embryonic kidney cell line 293 (HEK293) were obtained from the cell repository of the National Centre for Cell Science (NCCS), Pune, Maharashtra, India. Cell lines were authenticated by short tandem repeat (STR) profiling and tested for mycoplasma contaminations by the repository. HT1080 cell line was cultured with Dulbecco's Modified Eagle's Medium (DMEM, Himedia, Nashik, India; AL007A), and HEK293 cell line was cultured with Minimum Essential Medium Eagle (MEM, Himedia; AL047A). All the culture media were supplemented with 10% FBS (Gibco, Grand Island, NY, USA; 10270106) and 1% L-glutamine (Himedia; TCL012), and cells were cultured at 37 °C with 5% $CO_2$ in a humidified HeraCell 150i incubator.

**Establishment of DOX-resistant HT1080 cell sub-line**. For the development of DOX-resistant HT1080 cell sublines (HT1080/DOX), a stepwise selection method was used as described previously[13,63]. Initially, DOX-sensitive HT1080 cells were cultured with DMEM containing 10 nM DOX (Sigma, St. Louis, MO, USA;

D1515). When the cells were capable of growing in 10 nM DOX containing DMEM medium and reaching appropriate confluency, the cells were passaged, and DOX concentration was doubled. Further, the doubling of DOX concentration was carried out each time when the treated cells reached the growth rate of untreated cells. Finally, the DOX-resistant cell line, HT1080/DOX was established after five sequential treatments with a 300 nM final concentration of DOX. The resistance of HT1080/DOX cells to DOX was determined by measuring the $IC_{50}$ value and compared with HT1080 parental cells. The degree of DOX resistance was determined in terms of the resistant index ($R$)[13]. $R = IC_{50}$ of resistant cells/$IC_{50}$ of sensitive cells. The HT1080/DOX cells were maintained in DOX-free DMEM for at least two days prior to performing any further experiments.

**Transfection of piR-39980 mimic and inhibitor**. piR-39980 mimic and inhibitor were purchased from Integrated DNA Technologies (IDT, Coralville, IA, USA). piR-39980 mimic was designed using its mature sequence (DQ601914.1) and synthesized as a 2′ -O-methoxy modified RNA oligonucleotide. AllStars Negative Control siRNA (Qiagen, Hilden, Germany; 1027280) was used as a negative control for mimic (NC_Mimic) in this study. piR-39980 inhibitor and its negative control (NC_Inhibitor) were synthesized from IDT with N,N-diethyl-4-(4-nitronaphthalen-1-ylazo)-phenylamine ("ZEN") modification. The sequence of piR-39980 mimic, inhibitor, and negative control are listed in Supplementary Table 1. piR-39980 mimic/inhibitor and their corresponding control were transfected into HT1080 and HT1080/DOX cells using Lipofectamine-2000 (Invitrogen, Carlsbad, CA, USA; 11668027) and OptiMEM (Gibco; 31985070) following the manufacturer's instructions. Then, the cells were incubated for 6 h, after which the transfection mixture was replaced by complete DMEM and incubated for desired periods required for subsequent molecular functional assays.

**Small RNA isolation and qRT-PCR analysis**. To check the transfection efficacy, HT1080 cells were transfected with piR-39980 mimic/inhibitor or corresponding negative controls. Twenty-four hours after transfection, small RNAs were isolated using a mirVana miRNA isolation kit (Ambion, Austin, TX, USA; AM1560) following the manufacturer's protocol. The purity, quality, and concentration of small RNAs were measured using NanoDrop One instrument (Thermo Scientific) as per the manufacturer's instructions. The ratio of absorbance at 260 and 280 nm (260/280) and the ratio of absorbance at 260 and 230 nm were used to assess the purity of RNA. A 260/280 ratio of 1.8–2.0 and a 260/230 ratio in the range of 2.0–2.2 were accepted for pure RNA. The stability of RNA was assessed by agarose gel electrophoresis (Supplementary Fig. 11) as described previously[64,65]. RNA was converted into cDNA using the miScript PCR Starter Kit (Qiagen; 218193). The expression of piR-39980 upon mimic/inhibitor transfection was detected by qRT-PCR using the miScript SYBR Green PCR Kit (Qiagen; 218073) on the Quant-Studio 5 real-time PCR system (Applied Biosystem, Foster City, CA, USA; A34322). The expression of piR-39980 was normalized using small nucleolar RNA U6 as endogenous control. The primers used are synthesized from IDT (listed in Supplementary Table 2). All of the qRT-PCR assays were performed in triplicates, and the expression of piR-39980 was determined by the $2^{-\Delta\Delta Ct}$ method as reported previously[66].

To check the expression of piR-39980 in HT1080/DOX cells, we seeded ($2 \times 10^5$ cells/well) HT1080/DOX cells and HT1080 parental cells in a six-well tissue culture plate (Tarsons, Kolkata, India; 980010). After 24 h, cells were harvested, and qRT-PCR were performed as mentioned above. To investigate the effect of piR-39980 on DOX sensitivity in HT1080/DOX cells, we transfected the cells with piR-39980 mimic. The expression of piR-39980 upon mimic transfection was determined by qRT-PCR, as mentioned above.

**Determination of $IC_{50}$ value of DOX in HT1080 and HT1080/DOX cells**. The $IC_{50}$ value of DOX in HT1080 and HT1080/DOX cells was measured by 3-(4,5-dimethylthiazol-2-yl)−2,5-diphenyltetrazolium bromide (MTT) assay. HT1080 cells were seeded (50000 cells/well) in a 24-well tissue culture plate (Tarsons; 980030) and cultured for 24 h. Cells were treated with different concentration of DOX (0.1, 0.2, 0.3, 0.4, 0.5, 0.6, 0.7, 0.8, 0.9, and 1 μM) and incubated for 48 h. Then the media was discarded, and MTT (Sigma; M2003) was added to a final concentration of 0.5 mg/ml in incomplete DMEM. The cells were incubated for 4 h at 37 °C. The media was then discarded, and the cells were dissolved in DMSO (Himedia; TC185). The $IC_{50}$ value was determined by measuring the absorbance at 562 nm using a Multiskan SkyHigh Microplate Spectrophotometer (Thermo Scientific, Rockford, IL, USA).

The $IC_{50}$ value of DOX in HT1080/DOX cells was measured following the above procedure by treating the cells with 0.1, 0.5, 1.0, 1.5, 2.0, 2.5, 3.0, 3.5, and 4.0 μM DOX. We found 0.4 and 2.0 μM are the $IC_{50}$ value of DOX in HT1080 cells and HT1080/DOX cells, respectively.

**Cell viability assay**. Cell viability assay was performed initially to see the effect of piR-39980 on DOX sensitivity. HT1080 cells were seeded (50,000 cells/well) in a 24-well tissue culture plate (Tarsons; 980030) and cultured for 24 h. Then, the cells were transfected with 20 nM [concentration was selected from our previous study[29]] of mimic/inhibitor and the corresponding negative controls. After 24 h of transfection, cells were treated with 0.4 μM DOX and incubated for 48 h. Finally,

the relative cell viability was measured by conducting an MTT assay as mentioned above.

HT1080/DOX cells were resistant to 0.4 μM DOX. To increase the sensitivity of HT1080/DOX cells to 0.4 μM DOX, we transfected the cells with 20 nM mimic. The effect of piR-39980 on DOX sensitivity was determined by performing a cell viability assay as mentioned above.

**DOX-accumulation assay.** To investigate the effect of piR-39980 on DOX-accumulation within the HT1080 cells, HT1080 cells were seeded in a six-well tissue culture plate (Tarsons; 980010) at $2 \times 10^5$ cells/well and cultured for 24 h. Then, the cells were transfected with 20 nM mimic/inhibitor and the corresponding negative control. After 24 h of transfection, cells were treated with 0.4 μM DOX and incubated for 2 h. Then, cells were washed with 1X PBS and incubated with DMEM for another 4 h. After that, cells were washed with 1X PBS, and the accumulation of DOX within the cells was observed under an Epifluorescence microscope (Olympus IX71, Germany).

To quantify accumulated DOX within the cells, we seeded cells in a 96-well tissue culture plate (Tarsons; 980040) at 10,000 cells/well and incubated for 24 h. After piR-39980 mimic/inhibitor transfection and DOX treatment as mentioned above, the fluorescence intensity of accumulated DOX was measured using a multimode plate reader (Synergy H1 Hybrid Multi-Mode Microplate Reader, Biotek, USA) with excitation 470 nm and emission 595 nm.

To check the effect of piR-39980 on DOX-accumulation in HT1080/DOX cells, we transfected cells with 20 nM Mimic/NC_Mimic and treated them with 0.4 μM DOX. In addition, cells treated with only 0.4 μM and 2.0 μM DOX were included as an additional control group.

**PE Annexin-V apoptosis assay.** To determine whether piR-39980 boosts DOX-mediated apoptosis, we performed flow cytometry using PE Annexin-V Apoptosis Detection Kit I (BD, Franklin Lakes, NJ, USA; 559763). HT1080/DOX cells were seeded in a six-well tissue culture plate (Tarsons; 980010) at $2 \times 10^5$ cells/well and cultured for 24 h. Then, the cells were transfected with 20 nM mimic/inhibitor and corresponding negative controls. After 24 h of transfection, cells were treated with 0.4 μM DOX and incubated for 48 h. Then, the cells were harvested with 0.25% trypsin (Himedia; TCL007), washed twice with ice-cold 1X PBS, and resuspended in 500 μl binding buffer. About 5 μl Annexin-V-PE was added with the samples and incubated for 15–20 min in the dark. Cells were then analyzed using BD Accuri™ C6 Plus flow cytometry (BD, USA) within 1 h. The gating strategy and data processing is shown in Supplementary Fig. 12.

In addition, the HT1080/DOX cells were transfected with 20 nM mimic/NC_Mimic and treated with 0.4 μM DOX to check the effect of piR-39980 on DOX-induced cell death in DOX-resistant HT1080 cells.

**Colony formation assay.** HT1080/DOX cells were seeded ($1 \times 10^5$ cells/well) in a 12-well tissue culture plate (Tarsons; 980020) and transfected with 20 nM mimic or corresponding negative control. After 6 h of transfection, cells were treated with 0.4 μM DOX and incubated for 4 h. In addition, cells treated with only 0.4 μM and 2.0 μM DOX were included as an additional group. Then, cells were trypsinized and reseeded in a 60-mm Tissue Culture Petri Dish (1000 cells/dish) and cultured for 2 weeks. The culture media was replaced once every 3 days with 3 mL fresh DMEM containing 10% FBS. Then, the cells were washed with 1X PBS and fixed with 3.7% formaldehyde solution (Himedia; MB059) followed by staining with 0.1% crystal violet (Himedia; TC510) for 15 mins. The stain was washed three times with 1X PBS, and the colonies were counted.

In addition, we conducted a colony formation assay to check the effect of piR-39980 on the colony-forming ability of DOX-sensitive HT1080 cells. The HT1080 cells were transfected with 20 nM mimic/inhibitor and corresponding the negative controls. After 6 h of transfection, cells were treated with 0.4 μM DOX, and the colony formation assay was performed as described in the previous paragraph.

**Influence of piR-39980 on the morphology of DOX-treated cells.** To investigate the effects of piR-39980 on the cellular morphology of DOX-treated HT1080/DOX cells, cells were seeded in a six-well tissue culture plate (Tarsons; 980010) at $2 \times 10^5$ cells/well and cultured for 24 h. Then, the cells were transfected with 20 nM mimic/inhibitor and corresponding negative control. After 24 h of transfection, cells were treated with 0.4 μM DOX and incubated for 48 h. In addition, cells treated with only 0.4 μM and 2.0 μM DOX were included as an additional group. Cells were photographed under an inverted Epifluorescence microscope (Olympus IX71, Germany).

**Acridine orange/ethidium bromide (AO/EB) dual staining assay for apoptosis.** AO/EB dual staining assay was performed to detect apoptotic body formation in the apoptotic cells. This staining assay can discriminate among live cells, early apoptotic cells, and late apoptotic cells. After successful transfection with piR-39980 and treated with DOX as mentioned above, HT1080/DOX cells were washed with 1X PBS and incubated with 5 μg/mL (In 1X PBS) acridine orange (Himedia; MB116) and ethidium bromide (Sigma; E7637) at room temperature for 20 mins in the dark. Then the excessive stains were washed twice with 1X PBS, and the nuclei were observed under a confocal microscope (Leica TCS SP8). The AO was excited

with 502 nm wavelength light which emitted 525 nm wavelength green lights. The EB was excited with 526 nm wavelength light which emitted 605 nm wavelength orange-red light.

**Alkaline comet assay.** The effect of piR-39980 on DOX-induced DNA damage was measured by an alkaline comet assay. After completing piR-39980 transfection and DOX treatment in HT1080/DOX cells as mentioned in the above section, cells were washed with 1X PBS and harvested by trypsinization. After washing with 1X PBS, cells were resuspended in 500 μl 1X PBS. Then 200 μl of cell suspension was mixed with 800 μl of 1% low melting agarose (Himedia; MB080), having temperature 40 °C and spread onto a glass slide precoated with 1% agarose. The slides were incubated overnight at 4 °C in comet lysis buffer (1.2 M NaCl, 100 mM EDTA, 0.1% SDS, 0.26 M NaOH, pH >13). Then the slides were removed, washed with comet running buffer (0.03 M NaOH, 2 mM EDTA, pH >13) twice, and incubated for 20 min at room temperature in comet running buffer. Electrophoresis was conducted with a comet running buffer for 25 min in 20 volts. Then, the slides were removed from the electrophoresis chamber and neutralized in distilled water. The slides were stained with 2.5 μg/ml Propidium Iodide (Sigma; 81845) in distilled water for 20 min in the dark. The slides were rinsed with distilled water to remove excess stain. The comets were detected by an Epifluorescence microscope (Olympus IX71, Germany). The degree of DNA damage was evaluated by measuring comet tail length and tail moment using OpenComet (an automated comet assay image analysis tool available in the image processing platform, ImageJ)[67].

**γ-H2AX accumulation assay.** Phosphorylation of H2A histone family member X (H2AX) is the early cellular response to the induction of DNA double-strand breaks. Phosphorylated H2AX, termed γ-H2AX is a sensitive and high-throughput molecular marker for monitoring DNA damage initiation and resolution[33]. Therefore, we investigate γ-H2AX accumulation in the nucleus to measure the degree of DNA damage. HT1080/DOX cells were seeded in a six-well tissue culture plate (Tarsons; 980010) at $2 \times 10^5$ cells/well and cultured for 24 h. Then, the cells were transfected with 20 nM mimic/inhibitor or corresponding negative control. After 24 h of transfection, cells were treated with 0.4 μM DOX and incubated for 48 h. In addition, cells treated with only 0.4 and 2.0 μM DOX were included as an additional group. After that, we performed γ-H2AX accumulation assay following our previous protocol[28]. Rabbit monoclonal anti-gamma H2AX antibody (1:1000 dilution, Abcam, Cambridge, UK; ab81299) and DyLight 488 conjugated goat anti-rabbit secondary antibody (1:4000 dilution, Abcam; ab96899) were used in this study. The nuclei were observed under an Epifluorescence microscope (Olympus IX71, Germany). Three random fields were photographed, the nuclei contained γ-H2AX foci ≥4 were counted and plotted as Means ± SEM.

**piR-39980 target prediction.** The sequence of piR-39980 was retrieved from the GenBank (Accession no. DQ601914.1). We compiled a list of genes (CYP1A1, CYP1A2, ABCB1, GSTP1, MVP, EPHX1)[13], RRM1[68], RRM2[35], (ABCC3, ABCC6, JUNB, CLU)[69], (TOP2A, MCM4)[70] from previous studies on DOX resistance reported in different cancers such as breast cancer[13,69], adrenocortical cancer[68], pancreatic cancer[35], and gastric cancer[70]. The complete mRNA sequence of these genes was retrieved from NCBI. The target binding sites of piR-39980 on these genes was predicted using miRanda with the alignment score (SC) ≥170, binding energy (EN) ≤ −20.0 kcal/mol, strict Watson–Crick base pairing within primary seed site (2–11 nts at 5′ end of piRNA), and a less stringent base pairing within secondary seed site (12–21 nts at 5′ end of piRNA) tolerated with maximum three mismatches.

**qRT-PCR of target genes.** HT1080 and HT1080/DOX cells were seeded in six-well tissue culture plate (Tarsons; 980010) at $2 \times 10^5$ cells/well. After 24 h of incubation, cells were trypsinized and washed with 1X PBS twice. Total RNAs were isolated from both the cell lines using HiPurA™ Total RNA Miniprep Purification Kit (Himedia; MB602) according to the manufacturer's instructions. The quality and quantity of RNA were assessed as described previously in the section "small RNA isolation and qRT-PCR analysis". The cDNA was synthesized using RevertAid First Strand cDNA Synthesis Kit (ThermoFisher Scientific; K1622). The expression of predicted target genes was quantified by performing qRT-PCR using Hi-SYBr Master Mix (Himedia; MBT074) in QuantStudio 5 real-time PCR (Applied Biosystem, Foster City, CA, USA; A34322).

The expression of target genes was also measured by qRT-PCR after transfection of HT1080/DOX cells with 20 nM piR-39980 mimic/inhibitor and HT1080 cells with 20 nM piR-39980 mimic/inhibitor. RPL13 was used as an endogenous control for normalization, and data are expressed as $2^{-\Delta\Delta Ct}$. Primers were synthesized from IDT. Primers are listed in Supplementary Table 2.

**Dual-luciferase reporter assay.** The dual-luciferase reporter assays were conducted to check the direct physical interaction between piR-39980 and its target genes, CYP1A2 and RRM2. Partial wild-type (WT) sequences of 3′-UTR of RRM2 (NM_001165931, position 1176-1598) and CYP1A2 (NM_000761, position 2288-2630) containing piR-39980 binding site was cloned in between XhoI and NotI restriction sites in the 3′-UTR of Renilla luciferase gene of the psiCHECK-2 vector which contains both Renilla luciferase and firefly luciferase. The mutant constructs

(MUT) of *RRM2* and *CYP1A2* were generated in our laboratory by mutating fourth to eighth bases from 3′-end of the target site by site-directed mutagenesis (Supplementary Figs. 7 and 8). Primers are listed in Supplementary Table S2. The dual-luciferase reporter assay was performed by co-transfecting 50 ng WT/MUT-RRM2/CYP1A2 and 20 nM piR-39980 mimic using Lipofectamine-2000 reagent in HEK293 cells according to our protocol published earlier[29]. Renilla luciferase activity was calculated using the Dual-Luciferase Reporter Assay System (Promega, Madison, WI, USA; E1910).

**Construction of RRM2 and CYP1A2 overexpression vector and functional assays**. The coding sequence (CDS) along with partial 3′-UTR containing piR-39980 target site of *RRM2* and *CYP1A2* was amplified using 2X TaqMixture (Himedia; MBT061) in a T100 Thermal Cycler (Bio-Rad, Hercule, CA, USA) (Primers are listed in Supplementary Table 2). PCR products were cleaned and purified using HiPurA™ Quick Gel Extraction Kit (Himedia; MB539). Then the PCR products were digested with BamHI and XhoI restriction enzymes and cloned into pcDNA3.1(−) expression vector using Rapid DNA Ligation Kit (Thermo; K1422). Empty pcDNA3.1(−) was used as a control for the overexpression studies. We transfected pcDNA3.1 vector, pcDNA3.1-RRM2/CYP1A2 vector in HT1080 cells, and co-transfected pcDNA3.1-RRM2/CYP1A2 vector with piR-39980 using Lipofectamine-2000 reagent. Then we performed qRT-PCR of *RRM2* and *CYP1A2* to check its expression. Twenty-four hours after transfection, we treated the cells with 0.4 μM DOX. After 48 h incubation, we performed molecular assays such as MTT assay, DOX accumulation assay, PE Annexin-V apoptosis assay, comet assay, and γ-H2AX accumulation assay by adopting the protocols described above.

**Statistics and reproducibility**. The data organization, graphical representation, and statistical analysis were executed by GraphPad Prism 7.0. The data were represented in the graph as "mean ± SEM". All the in vitro experiments were performed three independent times in triplicates each time. Student $t$-tests were employed to compare the means between two groups, whereas one/two-way ANOVA was applied to compare the mean of three/more groups. $P < 0.05$ was considered statistically significant for all the data. *ns* – non-significant, * $P < 0.05$, ** $P < 0.01$, *** $P < 0.001$, **** $P < 0.0001$. The raw data for all the experiments are provided in the Supplementary Data 1 file.

**Reporting Summary**. Further information on research design is available in the Nature Research Reporting Summary linked to this article.

## Data availability

The source data underlying the graphs presented in the main figures are shown as Supplementary Data 1. All other data supporting the findings of the study are available within the paper and Supplementary Information.

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

## Author contributions

B.M. conceived and supervised the study. B.D. performed all the experiments. B.D. and N.J. analyzed the data with the help of B.M. B.D. and B.M. wrote the manuscript. All authors provided comments.

## Competing interests

The authors declare no competing interests.
