## [Transparent Peer Review File · Communications Biology]

Reviewers' comments:

Reviewer #1 (Remarks to the Author):

1. Please replace with a better clear picture of figure 6A and 8C.
2. Please adjust the color of Fig5C , Fig 8H, Fig 9E like Fig2F.
3. It's better to add the affect on Inhibitor of piR-39980 on HT1080 or HT1080/DOX cells.
4. Does Mimic and Inhibitor of piR-39980 affect the colony formation of HT1080?
5. Please add the influence of Inhibitor on RRM2 in HT1080/DOX in Fig 7B and the influence of Mimic on RRM2 in HT1080 in Fig 7C. Fig 7E,7F,7G and 7H have the same requirements.
6. Please add piR-39980/RRM2 or piR-39980/CYP1A2 regulates cell functions in HT1080/DOX cells.

Reviewer #2 (Remarks to the Author):

COMMSBIO-21-1419 :

The study "piRNA mediates doxorubicin resistance in fibrosarcoma by 3 regulating intracellular drug accumulation and DNA repair 4 through CYP1A2 and RRM2" deals with the investigation of the relationship between small ncRNAs called P-element-induced wimpy testis or PIWI-interacting RNAs (piRNAs), piR-39980 in particular, and doxorubicin (DOX) resistance in fibrosarcoma. The authors found that piR-39980 was less expressed in DOX-resistant HT1080 (HT1080/DOX) fibrosarcoma cells, and also its inhibition in parental HT1080 cells resulted in attenuated intracellular DOX accumulation, DOX-induced apoptosis and anti-proliferative effects. Overexpression of piR-39980 in HT1080/DOX cells promoted intracellular DOX accumulation, DNA damage and apoptosis, and negatively regulated the expression of RRM2 (a catalytic subunit of ribonucleotide reductase is crucial for DNA repair) and CYP1A2 (a major drug metabolizing enzyme). Further, this study suggested that the piRNA, piR-39980, can be used in combination with DOX for treatment of fibrosarcoma.

The idea and findings presented in the manuscript are novel and very interesting. Overall, the study was very well conducted and presented. The results obtained are very convincing and would be of significant relevance not only in the field of fibrosarcoma but also to a broader audience in cancer biology. However, there are few points that need to be addressed carefully prior to the consideration.

Comments:

1. For RT-qPCR, how is the RNA stability/integrity and concentration measured? The primers used for qRT-PCR in the study, should contain information on gene ID/accession number in the supplementary table if they are self-designed, or references should be provided if previously published. Also the reference for analysis of the gene expression is missing in the material and method sections.
2. The resolution for individual Flow Cytometry data in Figures 2E, 2F, 5C, 8H and 9E is very low. The authors need to enhance the resolution of these figures.
3. Lines 564-565, kindly include references for the reported studies of the genes involved in DOX resistance in different cancers.
4. Lines 716 and description in figure legends of figure 9, indicated the quantitation of intracellular DOX. However, lines 717-719 written as intercellular DOX. Kindly correct the errors.
5. Overall, the quality of the individual figures require enhancement in terms of quality, particularly those in figures 8 and 9.
6. The authors have done a wonderful work in executing their ideas with various experiments to prove and support their hypothesis in a very systematic manner.

Reviewer #3 (Remarks to the Author):

The growing interest in the role of ncRNAs in modulation of drug metabolism, gene expression pattern, and intracellular signaling has shed light into how various therapies can fail. The authors have interestingly explored the function of piR-39980 in increasing the sensitivity of DOX-resistant fibrosarcoma cells. In my point of view, this work has merits and the authors have supported their claim through sufficient data and lab approaches. However, there are some alterations recommended bellow which will improve the final draft.

1. In the study by Basudeb Das, (piR-39980 promotes cell proliferation, migration and invasion, and inhibits apoptosis via repression of SERPINB1 in human osteosarcoma. <https://doi.org/10.1111/boc.201900063>) the inhibition of piR-39980 induces apoptosis, whereas, in this study the authors claim that by the overexpression of piR-39980 they could resensitize DOX-resistant cells as well as inducing apoptosis. What are the reasons behind this variation? Is this difference due to the type of cancer or not? It would be appreciated if authors describe this difference in the discussion part as it will strengthen their claim and point of view.
2. Line 30; the sentence "Resistance to doxorubicin (DOX) is a significant barrier to treatment failure and tumor relapse in sarcoma" was rather confusing. How resistance can be a significant barrier to treatment failure and tumor relapse? I think this should be altered.
3. Lines 68 and 69; It would be better to provide solid evidence for the use of chemotherapy as the "best alternative" in sarcoma
4. It would be better if authors include solid references for piR-39980 target genes selected from other articles.

Responses to Reviewers # COMMSBIO-21-1419

We have successfully addressed all the comments raised by the reviewers, and the same has been incorporated in the revised manuscript. *All changes are marked up in RED in the marked-up version of the revised manuscript.* Please refer below for point-by-point responses to each of the comments and suggestions.

Reviewer # 1

1. Please replace with a better clear picture of figure 6A and 8C.

Response: As per your suggestion, we have now replaced the image. Please see *figure 6a, 8e, supplementary figure 4, and supplementary figure 9* of the revised manuscript.

2. Please adjust the colour of Fig 5C, Fig 8H, Fig 9E like Fig 2F.

Response: As per your suggestion, we have adjusted the colour of the flow cytometry figures. Please see revised *Figures 5c, 9b, and 10e*.

3. It is better to add the effect of Inhibitor of piR-39980 on HT1080 or HT1080/DOX cells.

Response: We have provided the data of the effect of inhibitor of piR-39980 on the cell viability, apoptosis, and DOX accumulation in HT1080 cells. In addition, we have now provided the data of the effect of inhibitor of piR-39980 on the colony-forming ability of HT1080 cells. Please refer to *Figures 1g, 2b, 2d, 3c, 3d, and 3f*.

4. Does Mimic and Inhibitor of piR-39980 affect the colony formation of HT1080?

Response: We would like to thank the reviewer for this suggestion, which has improved the quality of the manuscript. As per your suggestion, we have performed a colony-forming assay, and relevant texts are now incorporated in the revised manuscript. Kindly refer to marked-up texts in *Red* on *page 22* under the heading 'colony formation assay' (Methods), *page no 6* under the heading 'piR-39980 promotes sensitivity of fibrosarcoma cells to DOX' (Results), and *page 32 (legends of Fig. 2)*, and *figure 2a, 2b*.

5. Please add the influence of Inhibitor on RRM2 in HT1080/DOX in Fig 7B and the influence of Mimic on RRM2 in HT1080 in Fig 7C. Figs 7E, 7F, 7G, and 7H have the same requirements.

Response: We appreciate this fantastic suggestion of the reviewer. As suggested by the reviewer, we have performed qRT-PCR of RRM2 and CYP1A2 in HT1080, and HT1080/DOX cells upon transfection with piR-39980 mimic as well as inhibitor. The relevant changes are now incorporated appropriately in the revised manuscript. Kindly refer to marked-up texts in *Red* on *page 9 and 10* under the heading 'piR-39980 target RRM2 and CYP1A2' in the Results section and *figure 7a-f*. Further, inhibitor is not applicable to dual-luciferase reporter assay (*Figure 7g, h*). We have conducted the luciferase assay to confirm direct physical interaction between piR-39980 and its target gene. So, we have co-transfected RRM2/CYP1A2 overexpressing luciferase construct and piR-39980 mimic in HEK293 cells.

6. Please add piR-39980/RRM2 or piR-39980/CYP1A2 regulates cell functions in HT1080/DOX cells.

Response: We have explored the impact of RRM2/piR-39980 axis in DNA repair after DOX-induced DNA damage and the impact of CYP1A2/piR-39980 in DOX accumulation. We have

performed these experiments in DOX-sensitive HT1080 cells upon overexpression of RRM2 and CYP1A2. After that, we have co-transfected piR-39980 with RRM2/CYP1A2 overexpression vector to check whether piR-39980 rescue the effects or not (Figure 8, 9, and 10).

RRM2 and CYP1A2 are already highly overexpressed (figure 7a and 7d) in the DOX-resistance HT1080 (HT1080/DOX cells) cell line, and therefore, we did not overexpress RRM2/CYP1A2 further and performed these experiments in HT1080/DOX cells. To confirm that RRM2 and CYP1A2 induce chemoresistance we overexpress it in HT1080 parental cells.

RRM2/CYP1A2 showed ~10-fold upregulation in HT1080/DOX cells (figure 7a and 7d). Interestingly, when we transfected HT1080 parental cells with RRM2/CYP1A2 overexpression vector, we found ~10-fold upregulation of RRM2/CYP1A2 (Figure 8a and 10a). Therefore, we considered that upon overexpression of RRM2/CYP1A2 in parental HT1080 cells, these cells mimic their resistant counterpart. After overexpression of RRM2/CYP1A2, the IC₅₀ value of DOX in HT1080 cells was increased (Figure 8b and 10b). Thus, overexpression of RRM2/CYP1A2 in HT1080 parental cells induces DOX resistance, which is further rescued by piR-39980 mimic transfection (Figure 8, 9, and 10).

Reviewer # 2

1. For RT-qPCR, how is the RNA stability/integrity and concentration measured? The primers used for qRT-PCR in the study, should contain information on gene ID/accession number in the supplementary table if they are self-designed, or references should be provided if previously published. Also the reference for analysis of the gene expression is missing in the material and method sections.

Response: We appreciate this wonderful suggestion of the reviewer. As suggested by the reviewer, we have now provided the information regarding measurement of RNA integrity and concentration. Kindly refer to marked-up texts in Red on pages 19 and 24 under the heading 'Small RNA isolation and qRT-PCR analysis' and 'qRT-PCR of target genes' in the Methods section and supplementary figure 10.

The primers were self-designed. The GenBank ID of the genes is provided in Supplementary Table 2.

We have also added the reference for analysis of the gene expression analysis in the methods section. Kindly refer to the marked-up text in Red on page 19 under the heading 'Small RNA isolation and qRT-PCR analysis' in the methods section. The reference number is 66.

2. The resolution for individual Flow Cytometry data in Figures 2E, 2F, 5C, 8H, and 9E is very low. The authors need to enhance the resolution of these figures.

Response: As per your suggestion, we have improved the resolution of Flow Cytometry data. Kindly refer to revised Figures 2c, 2d, 5c, 9b, and 10e.

3. Lines 564-565, kindly include references for the reported studies of the genes involved in DOX resistance in different cancers.

Response: We appreciate this wonderful suggestion of the reviewer. As suggested by you, we now added references for all the genes selected from specific cancer. Kindly refer to marked-up texts in Red on page 24 under the heading 'piR-39980 target prediction' in the methods section.

4. Lines 716 and description in figure legends of figure 9 indicated the quantitation of intracellular DOX. However, lines 717-719 written as intercellular DOX. Kindly correct the errors.

Response: We would like to thank you for bringing this error to our notice. We have now corrected the error. Kindly refer to marked-up texts in Red on page 13 under the heading 'piR-39980/CYP1A2 modulates DOX-accumulation and apoptosis in fibrosarcoma' in the Results section.

5. Overall, the quality of the individual figures require enhancement in terms of quality, particularly those in figures 8 and 9.

Response: As per your suggestion, we improved all figures' quality, and they are now uploaded separately as image files to retain their quality.

Reviewer # 3

1. In the study by Basudeb Das, (piR-39980 promotes cell proliferation, migration and invasion, and inhibits apoptosis via repression of SERPINB1 in human osteosarcoma. <https://doi.org/10.1111/boc.201900063>) the inhibition of piR-39980 induces apoptosis; whereas, in this study the authors claim that by the overexpression of piR-39980 they could desensitize DOX-resistant cells as well as inducing apoptosis. What are the reasons behind this variation? Is this difference due to the type of cancer or not? It would be appreciated if authors describe this difference in the discussion part as it will strengthen their claim and point of view.

Response: We would like to thank the reviewer for this suggestion, which has improved the quality of the manuscript. As per your suggestion, we added a paragraph on different expressions and functions of piRNA in the discussion section. Kindly refer to marked-up texts in Red on page 17 under the 'Discussions' section.

2. Line 30; the sentence 'Resistance to doxorubicin (DOX) is a significant barrier to treatment failure and tumor relapse in sarcoma; was rather confusing. How resistance can be a significant barrier to treatment failure and tumor relapse? I think this should be altered.

Response: Suggestion has been incorporated. Kindly refer to marked-up texts in Red on page 2 under 'Abstract'.

3. Lines 68 and 69; It would be better to provide solid evidence for the use of chemotherapy as the 'best alternative' in sarcoma

Response: As per your suggestion, we have now added relevant references. Kindly refer to marked-up texts in Red on page 3.

4. It would be better if authors include solid references for piR-39980 target genes selected from other articles.

Response: As suggested by the reviewer, we have added references for all the genes selected from specific cancer. Kindly refer to marked-up texts in Red on page 24 under the heading '*piR-39980 target prediction*' of the Methods section.

REVIEWERS' COMMENTS:

Reviewer #2 (Remarks to the Author):

Revision Review Report
COMMSBIO-21-1419A :

In the revised version of the manuscript entitled "piRNA mediates doxorubicin resistance in fibrosarcoma by regulating intracellular drug accumulation and DNA repair through CYP1A2 and RRM2", the authors have done a wonderful job in improving with the presentation of their data and providing additional information to support their claims.

Overall, the authors have carefully answered in detail the comments that were pointed out, and corrected the errors and added missing information.

As per the revised version of the manuscript, I am very happy to recommend that this paper be published in this journal.

Reviewer #3 (Remarks to the Author):

In my opinion the authors have revised the issues which concerned me in the first place. Therefore I think this can be a potential manuscript to be published in the current version